# SATB2 induction of a neural crest mesenchyme-like program drives melanoma invasion and drug resistance

Maurizio Fazio[1,2†], Ellen van Rooijen[1,2†], Michelle Dang[1,2], Glenn van de Hoek[1], Julien Ablain[1,2], Jeffrey K Mito[1,3], Song Yang[1], Andrew Thomas[1], Jonathan Michael[1], Tania Fabo[1,2], Rodsy Modhurima[1,2], Patrizia Pessina[4], Charles K Kaufman[5,6], Yi Zhou[1,2], Richard M White[7], Leonard I Zon[1,2*]

[1]Howard Hughes Medical Institute, Stem Cell Program and the Division of Pediatric Hematology/Oncology, Boston Children's Hospital and Dana-Farber Cancer Institute, Harvard Medical School, Boston, United States; [2]Department of Stem Cell and Regenerative Biology, Harvard Stem Cell Institute, Cambridge, United States; [3]Brigham and Women's Hospital, Department of Pathology, Boston, United States; [4]Stem Cell Program and the Division of Pediatric Hematology/Oncology, Boston Children's Hospital and Dana-Farber Cancer Institute, Harvard Medical School, Boston, United States; [5]Division of Medical Oncology, Department of Medicine, Washington University in Saint Louis, Saint Louis, United States; [6]Department of Developmental Biology, Washington University in Saint Louis, St. Louis, United States; [7]Memorial Sloan Kettering Cancer Center and Weill-Cornell Medical College, New York, United States

*For correspondence:
zon@enders.tch.harvard.edu

†These authors contributed equally to this work

**Abstract** Recent genomic and scRNA-seq analyses of melanoma demonstrated a lack of recurrent genetic drivers of metastasis, while identifying common transcriptional states correlating with invasion or drug resistance. To test whether transcriptional adaptation can drive melanoma progression, we made use of a zebrafish mitfa:*BRAFV600E*;*tp53*-/- model, in which malignant progression is characterized by minimal genetic evolution. We undertook an overexpression-screen of 80 epigenetic/transcriptional regulators and found neural crest-mesenchyme developmental regulator SATB2 to accelerate aggressive melanoma development. Its overexpression induces invadopodia formation and invasion in zebrafish tumors and human melanoma cell lines. SATB2 binds and activates neural crest-regulators, including *pdgfab* and *snai2*. The transcriptional program induced by SATB2 overlaps with known MITF[low]AXL[high] and AQP1[+]NGFR1[high] drug-resistant states and functionally drives enhanced tumor propagation and resistance to Vemurafenib in vivo. In summary, we show that melanoma transcriptional rewiring by SATB2 to a neural crest mesenchyme-like program can drive invasion and drug resistance in autochthonous tumors.

## Introduction

Both sequencing of patient precursor lesions (*Shain et al., 2015*) and modeling in transgenic animals (*Pérez-Guijarro et al., 2017*; *van Rooijen et al., 2017*) showed that acquisition of oncogenic *BRAF* or *NRAS* mutations alone is insufficient to drive melanoma initiation. Reconstruction of the clonal history of advanced human melanoma suggests that few driver mutations, including loss of tumor suppressors, are acquired in early transformed melanocytes, and that high mutational burden and genomic instability are later events in malignant progression (*Birkeland et al., 2018*; *Ding et al., 2014*). Recently, a pan-cancer whole-genome analysis of metastatic solid tumors, including 248

melanoma patients, showed a surprisingly high degree of similarity in mutational landscape and driver genes between metastatic tumors and their respective primaries, as well as across multiple metastatic lesions within the same patient (*Priestley et al., 2019*). The lack of recurrently mutated genes in metastasis, and the limited genetic diversity across metastatic sites suggests that genetic selection of mutated drivers is unlikely to be responsible for metastatic progression. Meanwhile, despite the high degree of mutational burden and genetic intra- and inter-patient heterogeneity observed in melanoma, several bulk RNA-seq and scRNA-seq analyses of metastatic melanoma patient samples have identified common recurrent transcriptional states correlating with invasion (mesenchymal signature; *Verfaillie et al., 2015*), drug resistance (MITF$^{low}$/AXL$^{high}$; *Tirosh et al., 2016*), a neural crest-like state correlating with minimal residual disease persistence during targeted therapy with MAPK inhibitors (MITF$^{low}$/NGFR1$^{high}$/AQP1$^{high}$; *Rambow et al., 2018*), and even response to immunotherapy (NGFR1$^{high}$; *Boshuizen et al., 2020*). Yet, the existing literature on these often less proliferative cell states (*Rambow et al., 2018*) has been either descriptive (*Tirosh et al., 2016*), limited to in vitro perturbations, or relied on transplant models (*Rambow et al., 2018*; *Boshuizen et al., 2020*), and as such their role in oncogenesis outside of iatrogenic drug treatment remains less clear.

Using a zebrafish transgenic melanoma model, we have previously shown that in a cancerized field of melanocytes carrying driving oncogenic mutations, such as *BRAF*$^{V600E}$ and loss of *tp53,* only a few cells or subclones will go on to develop a malignant tumor, and that this event is marked by re-expression of developmental neural crest (NC)markers *sox10* and *crestin* (*Kaufman et al., 2016*; *McConnell et al., 2019*). Similar to the observation in human metastatic patient samples, malignant progression from transformed melanocytes in experimental animal models is not explained by the very limited genetic evolution observed in these tumors (*Yen et al., 2013*). Taken together, these observations in human patients and animal disease models raise the question whether epigenetic adaptation, rather than genetic selection, can contribute to tumor progression by altering the transcriptional state of the tumor.

Here, we address this hypothesis by high-throughput genetic perturbation of 80 epigenetic/transcriptional regulators in autochthonous primary tumors, by leveraging a transgenic zebrafish melanoma model uniquely suited to precisely and rapidly perturbate tumor development in vivo (*Ceol et al., 2011*), at a scale and speed much greater than possible with genetically engineered murine melanoma models. We identify Special AT-rich Binding protein 2 (SATB2) as a novel accelerator of melanoma onset driving an aggressive phenotype in primary tumors suggestive of metastatic spreading. SATB2 acts as a transcriptional regulator by recruiting members of the acetyl transferase and histone demethylase complexes to target genes and altering the local chromatin organization and activation state (*Zhou et al., 2012*). Its expression has been shown to correlate with patient outcome in several tumor types (*Naik and Galande, 2019*; *Yu et al., 2017b*). In different tissues and tumor types, SATB2 has been shown to play a role in oncogenic transformation and proliferation, or epithelial-to-mesenchymal transition (EMT), migration, and self-renewal (*Gan et al., 2017*; *Naik and Galande, 2019*; *Nayak et al., 2019*; *Wang et al., 2019*; *Wu et al., 2016*; *Xu et al., 2017*; *Yu et al., 2017a*). SATB2 is a transcription factor and chromatin remodeler with a well-conserved structure and expression pattern across chicken, mouse, and zebrafish during the development and migration of the cranial neural crest (CNC) and neuronal development. It is required for the development of the exo-mesenchymal lineages of the CNC (*Sheehan-Rooney et al., 2010*; *Sheehan-Rooney et al., 2013*), and neuronal axon formation (*McKenna et al., 2015*; *Shinmyo and Kawasaki, 2017*). In facts, SATB2 inactivating mutations in humans have been associated with cleft palate, intellectual disability, facial dysmorphism, and development of odontomas, defining a neurocristopathy referred to as SATB2-associated syndrome (*Kikuiri et al., 2018*; *Zarate et al., 2019*; *Zarate and Fish, 2017*). Through a combination of zebrafish in vivo allotransplants and validation in human melanoma cell lines, we show that SATB2 drives enhanced invasion via invadopodia formation and an EMT-like phenotype. Mechanistically, chromatin and transcriptional characterization of primary zebrafish SATB2 tumors vs. EGFP controls via ChIP-seq and RNA-seq shows SATB2 to bind and induce transcriptional activation of neural crest regulators, including *snai2* and *pdgfab*. The transcriptional program induced by SATB2 overexpression is conserved between zebrafish and human melanoma, and overlaps with the aforementioned MITF$^{low}$/AXL$^{high}$ (*Tirosh et al., 2016*) and neural crest-like MITF$^{low}$/NGFR1$^{high/}$AQP1$^{high}$ drug-resistant states (*Rambow et al., 2018*). Finally, we show SATB2

transcriptional rewiring to functionally drive enhanced tumor propagation and resistance to MAPK inhibition by Vemurafenib in zebrafish tumor allografts in vivo.

## Results

### In vivo overexpression screen of epigenetic factors identifies SATB2 as melanoma accelerator

To interrogate whether epigenetic reprogramming can accelerate melanoma development, we utilized a genetic discovery driven approach and undertook an in vivo overexpression screen. We tested 80 chromatin factors: 15 pools of 5 factors, and 6 additional single factors including previously published positive controls SETDB1, SUV39H1 (*Ceol et al., 2011*) (see *Supplementary file 1*). EGFP was used as a negative control, and we tested CCND1 as an additional positive control (known driver often amplified in melanoma [*Cancer Genome Atlas Network, 2015*]). As a screening platform, we leveraged a zebrafish melanoma model driven by tissue-specific expression of human oncogenic *BRAF^V600E* in a *tp53* and *mitfa*-deficient background (*Ceol et al., 2011*). Tg(*mitfa:BRAF^V600E*); *tp53-/-; mitfa-/-* zebrafish lack melanocytes and do not develop melanoma. Mosaic integration of the transposon-based expression vector MiniCoopR (MCR), rescues melanocyte development by restoring *mitfa,* while simultaneously driving tissue-specific expression of candidate genes (*Figure 1A*). Thus, in mosaic F0 transgenics all melanocytes express the candidate gene tested (described in *Ceol et al., 2011*). We identified six significant candidate accelerator pools (Pool B,D,F,G,I,L vs. MCR:EGFP Log-rank (Mantel-Cox) test p<0.0001****) of which Pool F had the strongest acceleration (*Figure 1B*), and tested individual factors from four accelerating pools (B/F/G/I) and a non-significant pool (C) (*Figure 1C* and *Figure 1—figure supplement 1A–D*). Single factor validation of Pool F (*Figure 1C*) showed that SATB2-overexpression (*Figure 1—figure supplement 2A–B*) is sufficient to strongly accelerate melanoma development (median onset of 12 weeks, compared to 21.4 weeks for MCR:EGFP, Log-rank (Mantel-Cox) test p<0.0001****) (*Figure 1C*). Single factor validation of additional pools (*Figure 1—figure supplement 1A–D*) identified additional genes, with a significant but milder acceleration of melanoma onset (TRIM28, median onset 18.7 weeks, n = 72; CDYL2, median onset 18.4 weeks, n = 72; DMAP1, median onset 18 weeks, n = 98; CBX5, median onset 19.1 weeks, n = 51; and PYGO2, median onset 19.7 weeks, n = 77), and CBX3 (median onset not reached,~20% of animals show tumors at 20 weeks, n = 55) to delay melanoma onset (*Figure 1—figure supplement 1D*). MCR:SATB2 tumors appear grossly aggressive (*Figure 1D*, *Figure 1—figure supplement 2C*, *Video 1*), are invasive, and melanoma cells are frequently observed in internal organs and spreading along the spinal cord, while organ involvement is rarely observed in MCR:EGFP (*Figure 1E–G*, *Figure 1—figure supplement 2C–F*). We thus focused on further characterizing SATB2's phenotype and investigating its mechanism of action. SATB2 effect on melanoma development is *BRAFV600E* and *tp53-/-*-dependent (*Figure 1—figure supplement 3A*). Melanocyte-specific expression of Cas9 and of a gRNA efficiently targeting zebrafish *satb2* (*Figure 1—figure supplement 3B*) using the established MCR:CRISPR system (*Ablain et al., 2018*) did not affect tumor onset compared to control targeting *tp53* (*Figure 1—figure supplement 3C*). While we did not verify *satb2* expression in transgenic tumors, this data suggest it to likely be dispensable for melanoma initiation. Furthermore, SATB2 closely related gene SATB1 did not affect tumor development (*Figure 1—figure supplement 3D*). The acceleration phenotype is not due to increased cellular proliferation, since SATB2-overexpression in primary zebrafish tumors (*Figure 1—figure supplement 4A*) or in a panel of human melanoma cell lines via a TETon tetracycline inducible lentiviral vector (here referred to as iSATB2) did not result in increased proliferation (*Figure 1—figure supplement 4B,C*). Our data demonstrate that SATB2-overexpression accelerates melanoma malignant progression without affecting proliferation, but SATB2 is not sufficient and unlikely to be required for melanoma initiation.

### SATB2 overexpression leads to invadopodia formation and increases migration and invasion in vitro and in vivo

Given the lack of proliferation changes, we next asked whether SATB2's aggressive tumor phenotype and internal tumors might be explained by an increase in tumor migration and/or invasion. Using primary zebrafish melanoma in vitro cell cultures, we performed scratch migration assays, that

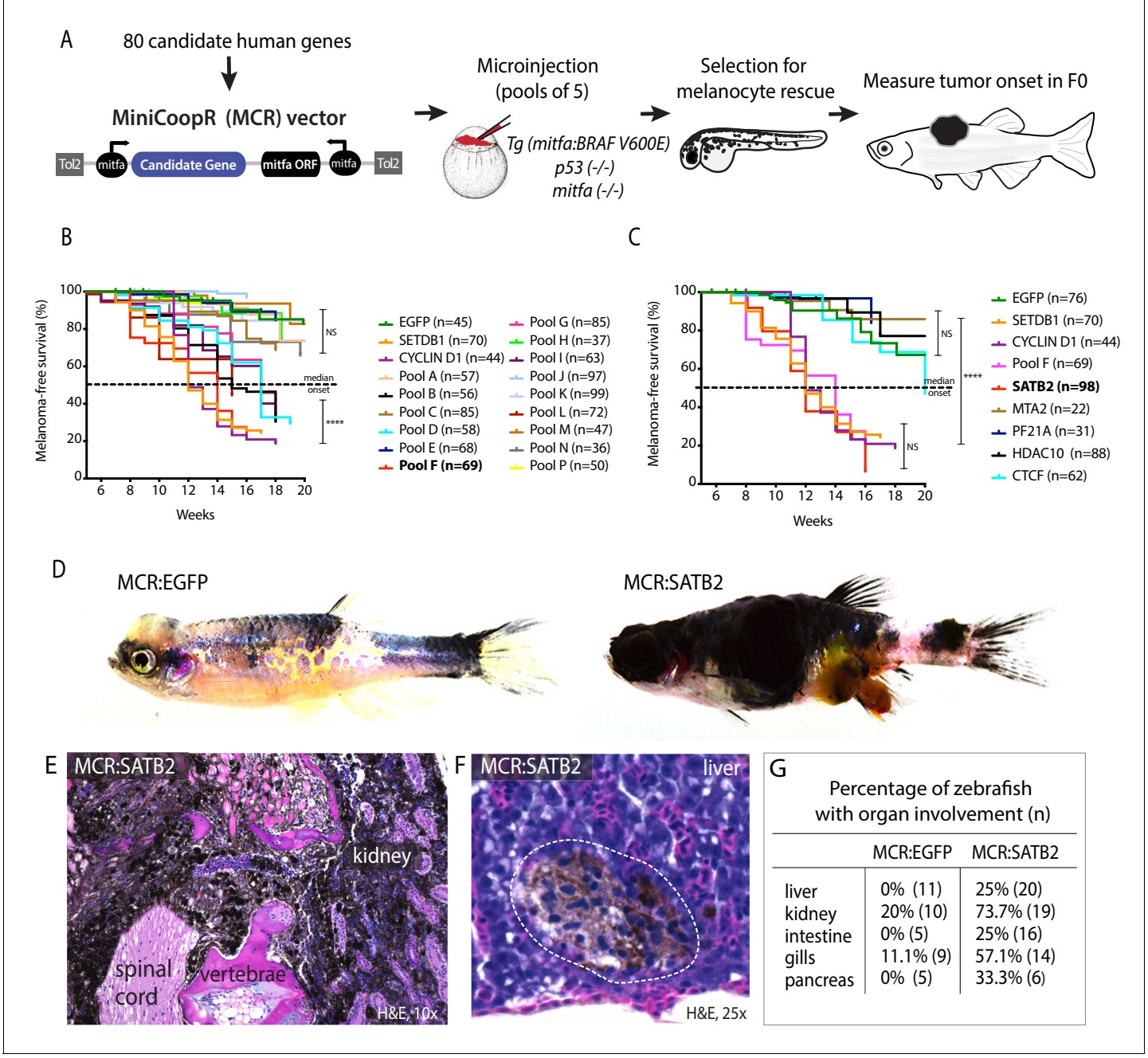

**Figure 1.** Overexpression screen of epigenetic regulators identifies SATB2 as an accelerator of melanoma formation in zebrafish. (**A**) Schematic overview of screening strategy: MCR expression vector-based reintroduction of the melanocyte master transcription factor *mitfa*, rescues melanocytes and melanoma development. All rescued melanocytes in F0 microinjected embryos also express a candidate human factor. (**B**) Kaplan-Meier melanoma-free survival curves of pooled chromatin factor screen. Six pools were significant [Log-rank (Mantel-Cox) test p<0.0001****]. Pool F (red) had the strongest acceleration effect (Median onset 14 weeks). (**C**) Single factor validation of pool F identifies SATB2 to induce accelerated melanoma onset [Median onset 12 weeks, Log-rank (Mantel-Cox) test, p<0.0001****]. (**D**) Zebrafish MCR:SATB2 tumors are aggressive compared to an MCR:EGFP age-matched control. (**E**) Histopathological analysis revealed that MCR:SATB2 tumors are highly invasive, here shown to invade through the spinal cord, vertebrae, and kidney. (**F**) Isolated melanoma cell clusters were found in the liver, and (**G**) frequent organ involvement is observed in MCR:SATB2 compared to MCR:EGFP controls.

The online version of this article includes the following source data and figure supplement(s) for figure 1:

**Source data 1.** Kaplan-Meier melanoma-free survival curves input files in weeks showing (1 = tumor, 0 = no tumor/censored, no value = lost at follow up).

**Figure supplement 1.** Single factor validation of additional pools.

*Figure 1 continued on next page*

*Figure 1 continued*

**Figure supplement 1—source data 1.** Kaplan-Meier melanoma-free survival curves input files in weeks showing (1 = tumor, 0 = no tumor/censored, no value = lost at follow up).
**Figure supplement 2.** SATB2 overexpression leads to invasive melanoma with organ involvement.
**Figure supplement 3.** SATB2 is not sufficient and unlikley necessary for melanoma initiation in zebrafish.
**Figure supplement 4.** SATB2 overexpression does not affect proliferation.

confirmed a heightened migration potential of 3 independent MCR:SATB2 cell lines compared to three independent MCR:EGFP cell lines (*Figure 2—figure supplement 1A*). Cytoskeletal staining of zebrafish melanoma in vitro cultures (Zmel1 MCR:EGFP and 45–3 MCR:SATB2) with phalloidin revealed the presence of strong F-actin-positive foci in MCR:SATB2 cells (*Figure 2A*), reminiscent of invadopodia. Invadopodia are cell protrusions involved in metastatic spreading by facilitating anchorage of cells to, and local degradation of the ECM (*Murphy and Courtneidge, 2011*), and are known to be regulated by the PDGF-SRC pathway and EMT. During normal development, invadopodia-like physiologically equivalent structures called podosomes are utilized by neural crest cells to migrate (*Murphy and Courtneidge, 2011*; *Murphy et al., 2011*). Co-localization of F-actin with invadopodium structural component Cortactin in MCR:SATB2 melanoma cell lines confirmed these foci to be invadopodia (*Figure 2A* and *Figure 2—figure supplement 1B*). Much like in vitro cultures, primary MCR:SATB2 tumors also showed abundant Cortactin expression compared to MCR:EGFP (*Figure 2B*). Proteolytic activity of invadopodia induces local degradation of the extracellular matrix. To test whether SATB2-expressing cells form functional invadopodia, we plated cells onto Oregon green 488-conjugated gelatin-coated coverslips and assayed for matrix degradation 24–25 hr post-seeding (*Martin et al., 2012*). MCR:SATB2 melanoma cell lines strongly degraded the gelatin matrix, with 49.2–57.9% of cells having degraded gelatin, compared to 5.4% of MCR:Empty control cells (*Figure 2C*, *Figure 2—figure supplement 1C–D*). To validate the effect of SATB2 as a regulator of invadopodia formation in human melanoma, we utilized the panel of iSATB2 human melanoma cell lines. Upon SATB2 induction (*Figure 1—figure supplement 4B*), all human melanoma cell lines robustly formed invadopodia and showed significantly increased matrix degradation (*Figure 2D–E*, *Video 2*). In SKMEL2, a 10-fold induction of cells with matrix degradation was observed, from a baseline of 5.4 ± 4.5% (SD) of cells with matrix degradation to 58.2 ± 13.2% (SD) of cells after doxycycline treatment (*Figure 2D–E*), with evidence of F-actin and Cortactin co-localization in punctae below the cell nucleus, characteristic of invadopodia (*Figure 2E*, *Figure 2—figure supplement 1E*).

To further validate whether the internal organ involvement in MCR:SATB2 primary tumors was due to an increased migratory potential in vivo, we used an orthotopic allotransplantation model in sub-lethally irradiated transparent *casper* zebrafish (*Heilmann et al., 2015*; *Li et al., 2011*; *White et al., 2008*). First, we allotransplanted zebrafish melanoma-derived cell lines Zmel1 (MCR: EGFP) vs. 45–3 (MCR:SATB2) into irradiated *casper* recipients. This showed SATB2 to cause an invasive histological phenotype and reduction in the recipient's overall survival compared to EGFP (45-3; n = 31, median survival = 25 days vs. Zmel1; n = 31, median onset not reached at the experimental

end point) (*Figure 2—figure supplement 2A–C*). We then tested whether this difference was also present in primary tumors using an established in vivo migration assay (*Heilmann et al., 2015*) where we transplanted 300,000 primary pigmented melanoma cells into *casper* recipients, and monitored the formation of distant metastasis (*Figure 2F*). One week post-transplantation, 18.6% (11/59) of MCR:SATB2 recipients already developed metastases while none were observed in MCR:EGFP recipients (0/27) (*Figure 2—figure supplement 2D*). At the experimental end point at 3–3.5 weeks-post transplantation, 59.4 ± 2.3% (SEM; n = 76 total recipients grafted from seven individual donor tumors) of MCR:SATB2 transplants formed

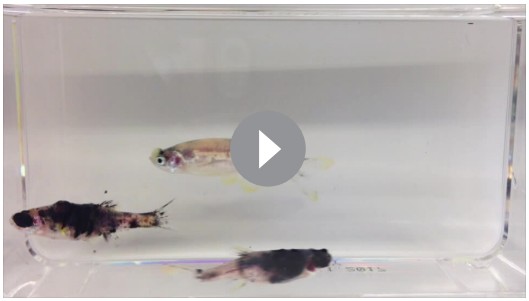

**Video 1.** Zebrafish MCR:EGFP (unpigmented) and MCR:SATB2 (pigmented) melanoma phenotype.
https://elifesciences.org/articles/64370#video1

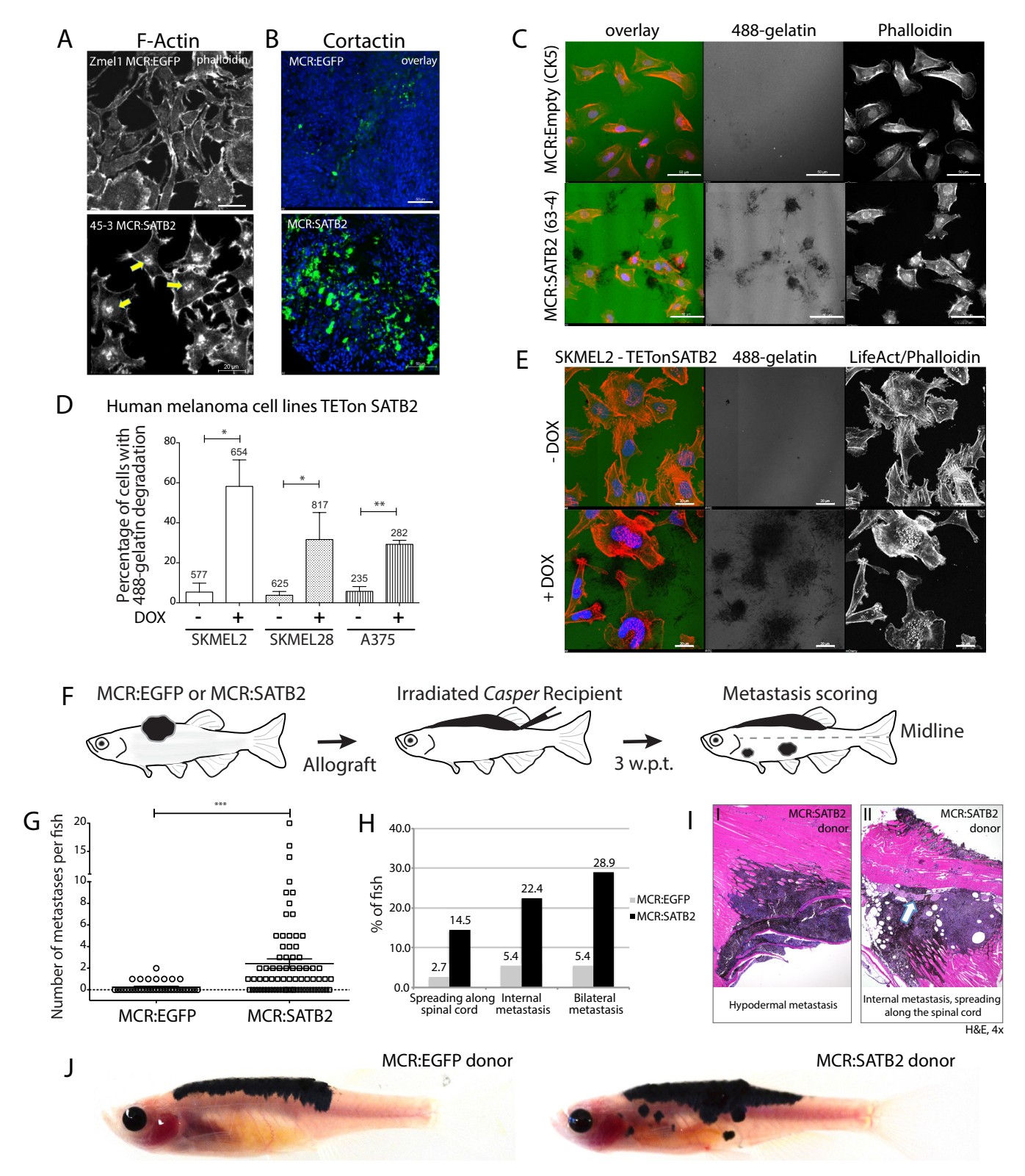

**Figure 2.** SATB2 leads to invadopodia formation and increased migration potential in vitro and in vivo. (**A**) Phalloidin staining for F-Actin in primary zebrafish melanoma cell culture, reveals the presence of F-actin-positive foci in MCR:SATB2 (45–3) cells, which are not present in MCR:EGFP (zmel1) cells. Scale bar is 20 μm. Primary tumor immunohistochemistry shows MCR:SATB2 tumors abundantly express Cortactin. Scale bar 50 μm. (**B**) MCR: SATB2 (63–4) cells show increased Oregon green 488-conjungated gelatin degradation, compared to MCR (CK5) cells 24–25 hr post-seeding. Scale bar

*Figure 2 continued on next page*

*Figure 2 continued*

is 50 μm. (C) Percentage of cells with degraded gelatin after SATB2 induction in iSATB2 human melanoma cell lines A375, SKMEL2, and SKMEL28 transduced with pInducer20-SATB2. Cells were seeded on gelatin in media +/- doxycycline after 48 hr +/- doxycycline induction. (D) Upon SATB2 induction in human melanoma cell line SKMEL2, cells form invadopodia and show increased matrix degradation Scale bar is 20 μm. (E) Orthotropic allograft migration assay in transparent *Casper* zebrafish. A total of 300,000 primary pigmented primary melanoma cells were transplanted into the dorsum of irradiated *Casper* recipients, which are monitored for the formation of pigmented distant metastasis that have spread past the anatomical midline. Metastases are represented as black circles. (F) Pooled recipient data at the experimental end point at 3.5 weeks post-transplantation, 59.4 ± 2.3% (SEM; n = 76 total recipients, grafted from seven individual donor tumors) of MCR:SATB2 transplants formed distant metastasis, compared to 21.8±4.5% (SEM; n = 37 total recipients grafted from five individual donor tumors) of EGFP-control transplants (p<0.0001). MCR:EGFP transplants developed an average 1.1 ± 0.4 (SD) distant metastasis per fish, versus 4 ± 4.2 (SD) in MCR:SATB2, where a maximum of 20 metastases per fish was observed. (G) MCR:SATB2 recipients more frequently developed bilateral (28.9%), and internal metastases (22.4%) compared to MCR:EGFP donors (5.4%), showing spreading along the neural tube (14.5% versus 2.7%). (H) At 3.5 weeks-post transplantation, compared to MCR:EGFP donor transplants, MCR:SATB2 transplants develop distant and internal metastases. Histopathology of MCR:SATB2 *Casper* recipients showing a (I–I) hypodermal metastasis, and (I–II) internal metastasis with spreading along neural tube.

The online version of this article includes the following figure supplement(s) for figure 2:

**Figure supplement 1.** SATB2 overexpression induces EMT, increased migration and invadopodia formation in zebrafish melanoma.
**Figure supplement 2.** MCR:SATB2 primary tumors and long-term cultures have increase invasion potential in vivo.
**Figure supplement 3.** Analysis of SATB2 expression, amplification, and correlation with survival in human melanoma.

distant metastasis, compared to 21.8 ± 4.5% (SEM; n = 37 total recipients grafted from five individual donor tumors) of EGFP-control transplants (two-tailed *t*-test p<0.0001) (*Figure 2G–J*, and *Videos 3–4*). In *casper* recipients that formed metastases, MCR:EGFP transplants developed an average of 1.1 ± 0.4 (SD) distant metastases per fish, versus 4 ± 4.2 (SD) in MCR:SATB2, where a maximum of 20 metastases per fish was observed (*Figure 2G*) (2-tailed *t*-test p<0.0001). Metastases most commonly spread to hypodermal sites (*Figure 2H–J*), similar to *in-transit* metastases that occur in human melanoma. MCR:SATB2 recipients more frequently developed bilateral metastases (28.9% versus 5.4% in MCR:EGFP), and internal metastases (22.4% versus 5.4%) which regularly spread along the neural tube (14.5% versus 2.7%) (*Figure 2H–I*). Collectively, these in vitro and in vivo data suggest that SATB2-overexpression induces invadopodia formation, increased migration and metastasis. SATB2 is endogenously expressed in human melanoma at the RNA (*Figure 2—figure supplement 3A*) and protein level (*Figure 2—figure supplement 3B*). Human melanoma patient genomic datasets publicly available on cBio portal show SATB2 to be infrequently but recurrently amplified in ~4–8% of patients (*Figure 2—figure supplement 3C*) in three independent datasets of metastatic melanoma (*Hugo et al., 2016*; *Snyder et al., 2014*; *Van Allen et al., 2015*), and it's high mRNA expression level correlate with poor survival in two independent metastatic melanoma patient datasets available on TIDE portal (GSE22153 and GSE8401, *Figure 2—figure supplement 3D*).

## SATB2 binds and transcriptionally regulates neural crest- and EMT-associated loci

To gain insight into the mechanism underlying the SATB2-overexpression phenotype in melanoma, we performed ChIP-seq and RNA-seq on primary zebrafish tumors. We conducted ChIP-seq on MCR:SATB2 primary zebrafish melanomas to identify SATB2-bound target genes. Of the panel of anti-SATB2 antibodies we tested (*Figure 2—figure supplement 3B* and *Figure 3—figure supplement 1A*) only SC-81376 worked for ChIP-seq. While the anti-SATB2 antibody SC-81376 cross-reacted with human SATB1 it did not recognize the endogenous zebrafish gene (*Figure 3—figure supplement 1A*), and as such should be specific for SATB2 in the MCR:SATB2 tumors. HOMER motif analysis

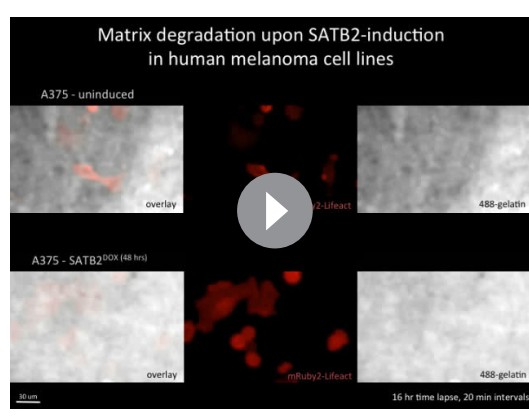

**Video 2.** Time-lapse video recording of Oregon Green 488-labeled gelatin degradation upon SATB2 induction in human melanoma cell line A375.
https://elifesciences.org/articles/64370#video2

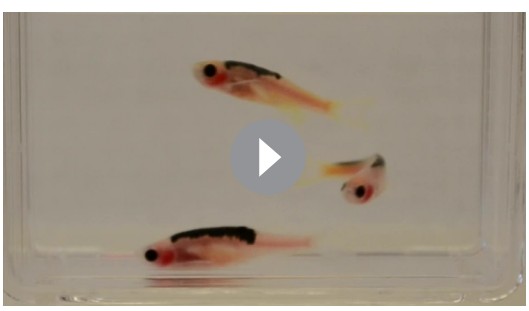

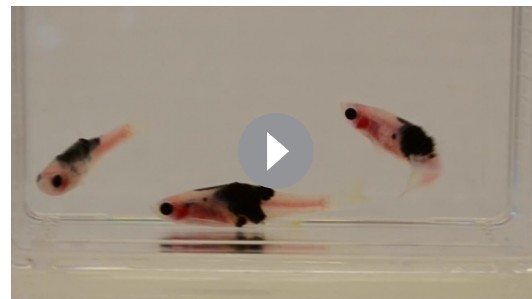

**Video 3.** MCR control primary tumor transplant into transparent *Casper* zebrafish, 3.5 weeks post-transplant. https://elifesciences.org/articles/64370#video3

**Video 4.** MCR:SATB2 primary tumor transplant into transparent *Casper* zebrafish, 3.5 weeks post-transplant. https://elifesciences.org/articles/64370#video4

of ChIP-seq significant peaks for de novo motif underlying SATB2 (Special AT-rich Sequence-Binding protein 2) binding showed the top motif to be AT-rich (*Figure 3—figure supplement 1B*) as expected (*Hassan et al., 2010*; *Naik and Galande, 2019*; *Savarese et al., 2009*; *Zhou et al., 2012*). GREAT analysis of GO-terms associated with ChIP-seq peaks, showed SATB2 binding to be enriched at loci associated with NC development and migration, and EMT (*Figure 3A*). Indeed, ~38.8% (127/327) of known NC-associated genes (*Tan et al., 2016*), including *sox10, snai2, pax7, chd7, and semaforin* (*sema3fa*) are bound by SATB2 (*Supplementary file 2*). We next performed ChIP-seq for H3K27Ac and H3K9me3 histone marks in MCR:SATB2 and MCR:EGFP control tumors to investigate the effect of SATB2 overexpression on the chromatin state of SATB2-bound targets. Globally, genome-wide GO term analysis of loci with increased H3K27Ac deposition in MCR:SATB2 vs. MCR:EGFP also showed an increased activation of neural crest development (*Figure 3—figure supplement 1C*). Furthermore, HOMER motif analysis of known motifs shows SATB2 ChIP-seq to be enriched for TFAP2A and RXR motifs, which are both transcription factors involved in neural crest specification (*Figure 3—figure supplement 1D*). Locally, H3K27Ac deposition in MCR:SATB2 vs. MCR:EGFP around transcriptional start sites (TSS) of SATB2-bound genes suggested an increased chromatin activation state of SATB2-bound NC targets (*Figure 3B*).

SATB2's binding pattern is consistent with its role as a necessary specifier of cranial migratory neural crest differentiation in exo-mesenchymal tissues in the pharyngeal arches (which will develop into the jaw and teeth), and the consequent cleft palate defects observed in patients with mutated *SATB2* in the human SATB2 syndrome (*Zarate and Fish, 2017*). Consistent with these prior literature findings, *satb2* is highly expressed in a published dataset of migrating *crestin:EGFP*+-NC cells sorted from zebrafish embryos at the 15 somite stage (*Kaufman et al., 2016*), when EMT-mediated delamination initiates migration of the neural crest, including cranial-NC (*Figure 3—figure supplement 1E*). Furthermore, knock-down of *satb2* during embryonic development via microinjection of a validated splicing morpholino (*Ahn et al., 2010*) in the transgenic neural crest reporter zebrafish line Tg(*sox10:mCherry*) showed a reduction of *sox10* reporter expression and craniofacial abnormalities, but did not affect melanocyte development (*Figure 3—figure supplement 1F*).

To define the transcriptional effect of SATB2-overexpression we performed RNA-seq on 3 MCR:SATB2 and 3 MCR:EGFP tumors, and correlated SATB2-bound loci with their transcriptional changes in MCR tumors to identify SATB2 target genes that might underlie SATB2's phenotype (*Figure 3C*). Given the inter-tumor transcriptional variability of primary zebrafish tumors (evident by qPCR on SATB2 itself in MCR:SATB2 tumors, *Figure 3D*), we conducted extended validation on a subset of these bound and transcriptionally altered genes, plus some additional manually curated SATB2-bound genes by qPCR on a large set of primary tumors (*Figure 3D*). Genes that are SATB2-bound (within 3 kb of the transcriptional start or end site, and the gene body) or SATB2-associated (predicted nearest gene association), which were altered by RNA-seq revealed that NC, cytoskeleton and extracellular matrix-associated gene programs are highly stimulated by SATB2-overexpression (*Figure 3D*, *Supplementary file 2*). *chd7* and *snai2* in particular have been described to be key regulators of the migratory NC (*Bajpai et al., 2010*; *Okuno et al., 2017*; *Schulz et al., 2014*), while *pdgfab* has been implicated in podosome and invadopodia formation (*Ekpe-Adewuyi et al., 2016*; *Murphy and Courtneidge, 2011*; *Murphy et al., 2011*; *Paz et al., 2014*). SATB2-binding to targets

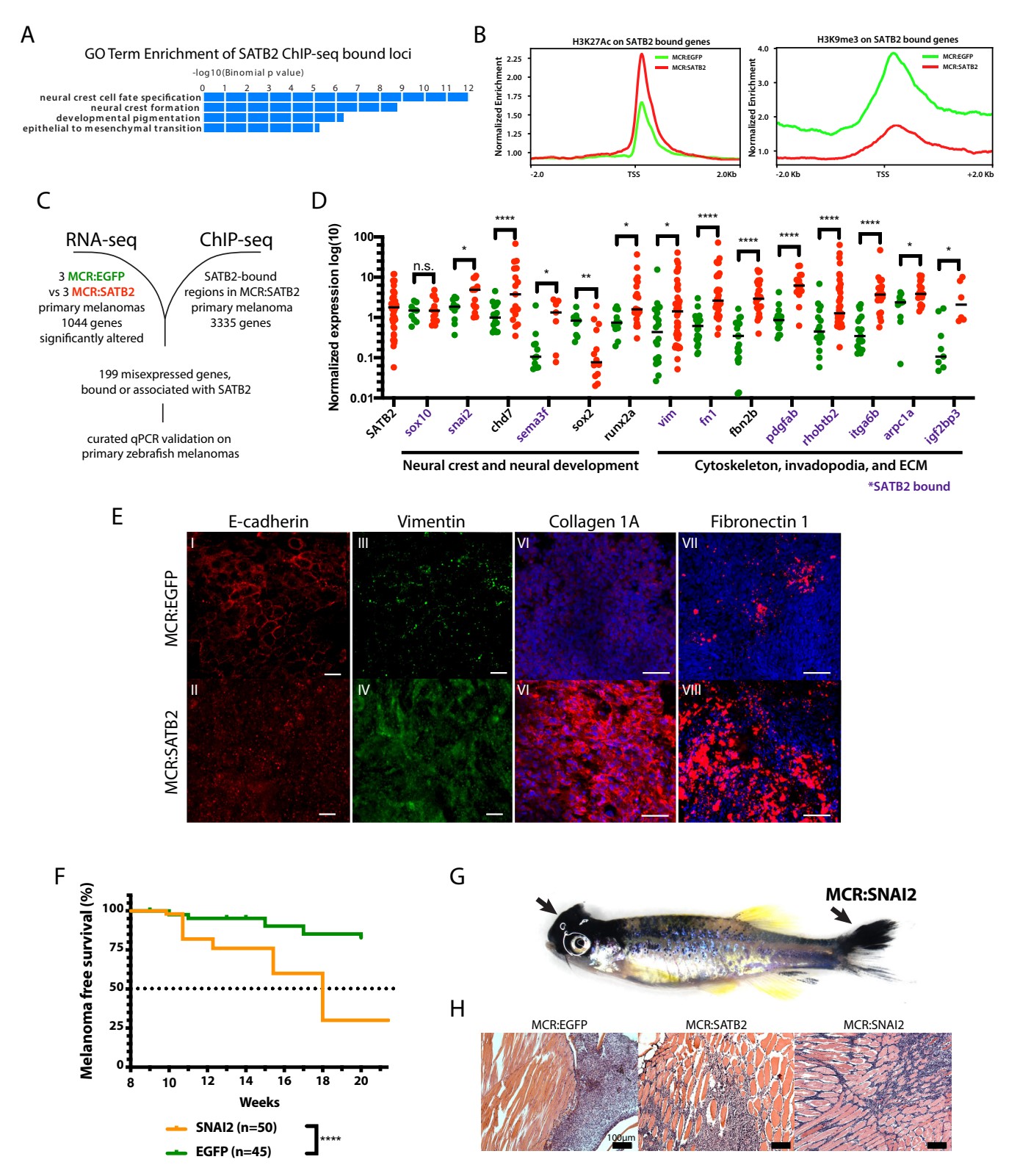

**Figure 3.** SATB2 binds and regulates EMT and neural crest-associated genes. (A) GO-term enrichment of SATB2-bound loci (GREAT analysis) from anti-SATB2 ChIp-seq on MCR:SATB2 tumors. Genes are defined as bound if SATB2 peaks are found within 3 kb of the transcriptional start (TSS) or end site (TES), and the gene body (GB). (B) H3K27Ac and H3K9me3 TSS +/- 2 Kb profile of H3K27Ac at SATB2 bound targets comparing MCR:SATB2 in red and MCR:EGFP in green. (C) ChIP-seq and significant differentially expressed genes RNA-seq ($p<0.05$, $q < 0.05$, FC>|1.5|) overlap was performed on primary

*Figure 3 continued on next page*

Figure 3 continued

zebrafish tumors to identify SATB2-bound genes that are misregulated. (D) qRT-PCR validation of transcriptionally altered SATB2-bound and associated targets associated to neural crest development, and the actin cytoskeleton and extracellular matrix. Symbols indicate single primary tumors, normalized to β-actin. *p<0.05, ***p<0.0001, ****p<0.0001. SATB2-bound genes are highlighted in purple. (E) Whole mount primary tumor immunohistochemistry for E-cadherin (red), scale bar is 100 μm, Vimentin (green), scale bar is 100 μm, Fibronectin 1 (red), scale bar is 50 μm, and Collagen 1 (red), scale bar is 50 μm, Dapi (blue). (F) Injection in MCR model as described in *Figure 1A* of MCR:EGFP, and MCR:SNAI2. (Log-rank (Mantel-Cox) test p<0.0001****). (G) Gross anatomy of zebrafish injected with MCR:SNAI2. Arrows show melanomas. (H) H and E representative histology of MCR:EGFP, MCR:SATB2, and MCR:SNAI2 tumors. Black bar is 100 μm.

The online version of this article includes the following figure supplement(s) for figure 3:

**Figure supplement 1.** Analysis of SATB2 expression and chromatin effect in human and zebrafish melanoma.

such as *pdgfab* (*Figure 3—figure supplement 1G*), results in increased H3K27Ac deposition along the promoter, TSS region and gene body in MCR:SATB2 compared to MCR:EGFP, respectively.

To investigate whether in MCR:SATB2 tumors cell fate rewiring functionally induces an EMT-like phenotype-switch reminiscent of the CNC development, we performed whole mount immunofluorescence for EMT markers on primary zebrafish melanomas. Indeed, compared to MCR:EGFP, MCR:SATB2 tumors showed mislocalized E-cadherin, and an increased protein expression of Vimentin and extracellular matrix attachment proteins Fibronectin 1 (*fn1a/b*) and Collagen 1 (*col1a1a/b*) (*Figure 3E*). Given the known role of *snai* family of proteins in regulating EMT we tested whether SATB2's direct downstream target *snai2* could recapitulate the MCR:SATB2 phenotype in vivo. Overexpression of SNAI2 resulted in a significant acceleration of tumor onset compared to the EGFP negative control (*Figure 3F*), albeit milder than SATB2 (MCR:SNAI2 median onset 18 weeks vs 12 weeks for MCR:SATB2). Similar to MCR:SATB2 (*Figure 1D*), and unlike MCR:EGFP, MCR:SNAI2 fish showed development of multiple tumors (*Figure 3G*), with a histopathologically invasive morphology (*Figure 3H*). Overall, SATB2 binds and activates chromatin at a subset of neural crest-related loci including *snai2*, resulting in an EMT-like phenotype. Overexpression of the human ortholog of downstream target *snai2* partially recapitulates SATB2's phenotype in the MCR model.

## SATB2-induced program is conserved across zebrafish and human melanoma and overlaps with known drug resistance transcriptional states

To ascertain whether SATB2-induced transcriptional changes were conserved across species, we induced SATB2 overexpression in primary human melanocytes and human melanoma cell lines, and performed qPCR for selected orthologs of genes validated in zebrafish. qPCR analysis of SATB2 target genes after 48 hr of culture in the presence of doxycycline on iSATB2 melanoma cell lines (SKMEL2, A375, and SKMEL28 iSATB2), and iSATB2 untransformed primary human melanocytes (HEMA-LP) showed a similar increase in SNAI transcription factors (*Figure 4A*). Cell-line-specific differences in overall *SNAI1* or *SNAI2* (both *snai2* orthologs) induction levels may reflect context dependence (e.g. oncogene or cell line baseline status) (*Nieto et al., 2016*). To analyze global transcriptional changes induced by SATB2 in human melanoma, we selected SKMEL2 iSATB2, since this cell line had the lowest endogenous SATB2 level and showed the strongest invadopodia induction (*Figure 2D–E* and *Figure 2—figure supplement 1E*). We conducted RNA-seq on SKMEL2 iSATB2 after 48 hr of treatment with doxycycline. Both Gene Set Enrichment Analysis (GSEA) Hallmark pathways analysis and Ingenuity Pathway Analysis (IPA) (*Figure 4B*) of significant genes, showed similar effects of SATB2 overexpression between the human melanoma cell line (SKMEL2 iSATB2 p<0.01; q < 0.01; FC > 2) and zebrafish tumors (MCR:SATB2 vs. EGFP p<0.01; q < 0.01; FC > 1.5) with strong similarity in the top significant pathways across human and zebrafish melanoma. These overlaps included pathways related to EMT/mesenchymal activation, osteoblast changes, axonal guidance, semaphorins, cancer metastasis, WNT, and TGFB signaling (*Figure 4B*). SATB2-overexpression (SKMEL2 iSATB2 RNA-seq) induced consistent transcriptional changes in several NC and invadopodia regulators, including activation of human orthologs of *pdgfab* and *snai2* (*Figure 4C*). These similarities across species, overexpression systems, and in vitro vs. in vivo conditions consolidate our findings.

To asses how the SATB2 program relates to previously described transcriptional states in melanoma, we conducted GSEA analysis in SKMEL2 iSATB2 for known signatures: (1) a mesenchymal

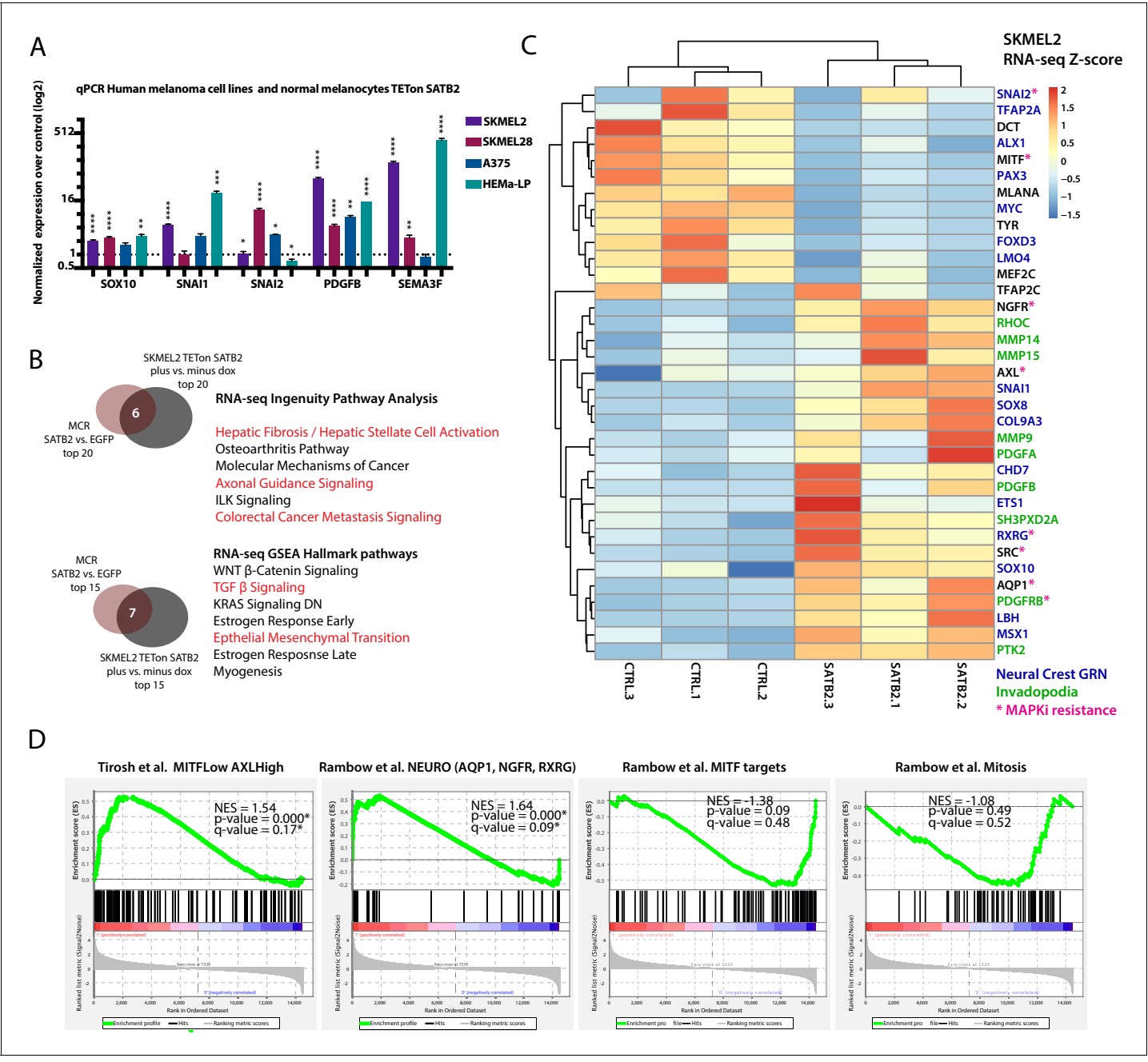

**Figure 4.** Conservation between transcriptional effects induced by SATB2 in zebrafish and human melanoma, and between the SATB2-program and known drug-resistant cell states. (**A**) Overexpression of SATB2 with TETon (iSATB2) in normal human epidermal melanocytes (HEMA-LP) and in SKMEL2, SKMEL28, and A375 melanoma cells is sufficient to induce transcriptional activation of *SOX10, SNAI1/2, PDGFB,* and SEMA3F mRNA. qRT-PCR Data was normalized to beta-actin, and fold change was determined compared to un-induced controls. *p<0.05, **p<0.001, ***p<0.0001, ****p<0.0001. (**B**) Overlap of Ingenuity Pathway Analysis and Gene Set Enrichment Analysis Hallmark pathways of RNA-seq in MCR zebrafish tumors and iSKMEL2 human (plus vs. minus doxycycline) melanoma cell line. For IPA humanized significant zebrafish genes were used (p<0.05, q < 0.05, FC|1.5|=1044 zf->840 unique human orthologs), and significant SKMEL2 iSATB2 genes (p<0.05, q < 0.05, FC>|1|). (**C**) Curated heatmap of altered genes in RNA-seq of human SKMEL2 cells overexpressing SATB2 (plus DOX vs. minus DOX FC >1; p<0.01) that have been involved in neural crest or Rambow Neural crest-like state, PDGF-PDGFR-SRC invadopodia cascade, AXL/MITF state and melanocyte differentiation. All genes shown are significantly altered except for TFAP2C. Normalized RNA-seq heatmap of expression fold change Z-score is plotted. Genes part of the neural crest GRN are highlighted in blue, invadopodia related genes are highlighted in green, and MAPK inhibitors resistance genes are highlighted in magenta. (**D**) Gene set enrichment analysis (GSEA) of 1000 MSigDB signatures including the states described by Tirosh et al. and Rambow et al.

The online version of this article includes the following figure supplement(s) for figure 4:

**Figure supplement 1.** IPA analysis of significant differentially expressed genes in SKMEL2 iSATB2.

signature (*Verfaillie et al., 2015*), (2) MITF$^{low}$/AXL$^{high}$ drug resistance state (*Tirosh et al., 2016*), and (3) neural crest state and other transcriptional signatures in different tumor cell subpopulations (*Rambow et al., 2018*). This analysis showed a significant overlap of the SATB2-induced transcriptional program with the less differentiated MITF$^{low}$/AXL$^{high}$ state (NES = 1.54, p=0.000, *q* = 0.17), and the Rambow neural crest-like/minimal residual disease MITF$^{low}$/NGFR1$^{high}$/AQP1$^{high}$ state driven by *RXRG* (NES = 1.64, p=0.000, *q* = 0.09) (*Figure 4D*), which have both been previously described in metastatic human melanoma to correlate with a self-renewing MAPK inhibitor drug-resistant state (*Rambow et al., 2018*; *Tirosh et al., 2016*). The key marker genes of these states (i.e. *AQP1, RXRG, NGFR, AXL, Figure 4C*) and additional genes involved in MAPK resistance (i.e. SRC and PDGF pathway members (*Ekpe-Adewuyi et al., 2016*; *Nazarian et al., 2010*; *Rebecca et al., 2014*), *Figure 4C*) and RXR-related pathways (*Figure 4—figure supplement 1A–B*) are significantly induced by SATB2 overexpression. Overall, SATB2 drives a transcriptional induction of invadopodia, EMT and neural crest regulators in zebrafish and human melanoma alike, consistent with the transcriptional changes and phenotypes described above and its known role in development. Furthermore, the SATB2-induced transcriptional program shows strong overlaps with known drug-resistance transcriptional states in melanoma.

## SATB2 increases tumor propagating potential and resistance to MAPK inhibition in vivo

Given the overlap between the SATB2-induced program and the known minimal residual disease and MAPK pathway inhibition-resistant states, we asked whether SATB2 functionally confers enhanced self-renewal or resistance to MAPK pathway inhibition. To address the former, we performed a limiting dilution tumor propagation assay injecting 300,000, 3000, 300, or 30 primary pigmented melanoma cells from MCR:EGFP (n = 4) and MCR:SATB2 (n = 5) donor tumors, into the dorsum of the zebrafish (*Figure 5A*). While MCR:EGFP did not display engraftment with fewer than 3000 cells (0/28), 44% (4/9) of MCR:SATB2 engrafted with 30 cells (*Figure 5B*). SATB2 donors have a significantly higher estimated fraction of tumor propagating cells (median estimate 1/21.6 cells for MCR:SATB2 vs. 1/10,879 cells for MCR:EGFP, Mann Whitney p=0.0159*; *Figure 5C*).

To test whether the SATB2-induced program confers resistance to MAPK pathway inhibition, we utilized an established in vivo drug treatment assay (*Dang et al., 2016*) where primary zebrafish melanoma are allotransplanted in irradiated *casper* recipients and administered a BRAFi (Vemurafenib 100 mg/kg) daily via oral gavage starting at day 10 post transplant (*Figure 5D*). Analysis of tumor growth by measuring tumor area with digital calipers after 2 weeks (day 24 post-transplant) of treatment with Vemurafenib or a DMSO vehicle control showed a complete lack of response in MCR:SATB2 compared to Vemurafenib sensitive MCR:EGFP tumors (*Figure 5E–F*) (MCR:EGFP BRAFi vs. MCR:SATB2 BRAFi 2-tailed *t*-test p<0.0001****). Our data functionally validate SATB2 as a driver of enhanced tumor propagation and drug resistance in vivo.

## Discussion

In this study, we investigated the effect of epigenetic and transcriptional regulators on melanoma development, by in vivo perturbation in endogenous tumors. We screened 80 human chromatin factors using a pooled screening approach in a transgenic zebrafish melanoma model driven by the most common melanoma oncogene, *BRAF$^{V600E}$*, and loss of the tumor suppressor *tp53* (*Cancer Genome Atlas Network, 2015*). This experimental approach could be used to try to identify pools/genes that can accelerate (e.g. SATB2, *Figure 1C*) or delay (e.g. CBX3, *Figure 1—figure supplement 1D*) tumor onset. Both for biological and for technical reasons we chose to focus on accelerators including: (1) A longer screen follow up time is required to identify pools/genes slowing tumor onset and (2) the likelihood of false positives in delayed tumor onset from perturbation resulting in development toxicity is higher. Through single factor validation of pool F, which had the strongest acceleration phenotype, we identified transcriptional regulator SATB2 as a potent accelerator of melanoma onset in zebrafish (*Figure 1B–C*, *Video 1*). Of the other six significant genes identified in the screen (TRIM28, CDYL2, DMAP1, CBX5, PYGO2, *Figure 1—figure supplement 1A–D*), TRIM28 is known to increase melanoma tumor propagation potential (*Czerwinska et al., 2020*).

MCR:SATB2 transgenic animals, in addition to developing tumors faster, had a high prevalence of tumor burden in internal organs, which is uncommon in MCR:EGFP controls, even in animals with

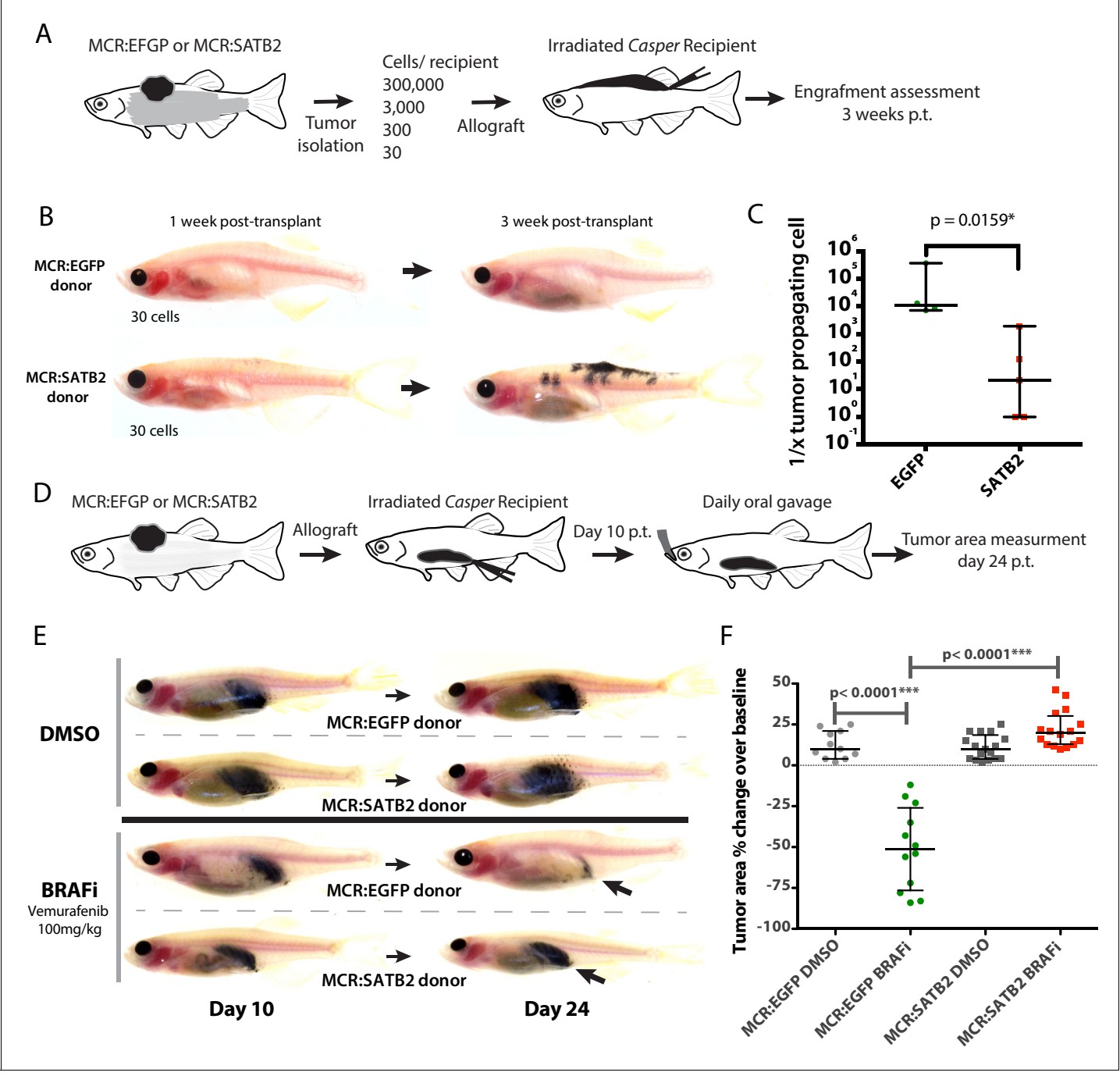

**Figure 5.** MCR:SATB2 tumor allografts have higher tumor propagating potential and resistance to Vemurafenib in vivo. (**A**) Weekly time lapse photography of a representative MCR:SATB2 melanoma cell recipient. Black arrows indicate metastases, which can already be observed 1 week post-transplantation. Related to *Figure 2E–I*. (**B**) Subcutaneous primary melanoma limiting dilution transplantation model in transparent *Casper* zebrafish. 300,000, 3000, 300, or 30 primary pigmented primary melanoma cells were transplanted into a subcutaneous space in running along the dorsum of irradiated *Casper* recipients. Ten to 12 animals per condition were used, and a total of four MCR:EGFP and five MCR:SATB2 donors were transplanted over the course of two independent experiments. (**C**) Representative images of MCR:EGFP and MCR:SATB2 recipients transplanted with 30 cells. (**D**) Engraftment of surviving recipients at 3 weeks post-transplant was used to estimate the frequency of tumor propagating cell potential with Extreme Limiting Dilution Analysis (ELDA). Estimated frequency of tumor propagating cells between MCR:EGFP and MCR:SATB2 donors (Mann-Whitney, p<0.0159*). (**D**) In vivo drug treatment of allotransplanted primary MCR:EGFP or MCR:SATB2 tumors with FDA-approved BRAF inhibitor Vemurafenib shows confirms SATB2 as a resistance driver. Recipients casper fish were treated by daily oral gavage with 100 mg/kg of drug or vehicle control from day 10 post-transplant to day 24 post-transplant. Animals were imaged at day 10 and 24 and pigmented tumor area was estimated using digital calipers to assess treatment response. (**E**) Representative image of individual treated animals. (**F**) Mean normalized area % change in the treated

*Figure 5 continued on next page*

Figure 5 continued

cohorts: MCR:EGFP DMSO (n = 11) +12.45, SEM 2.55 vs. MCR:EGFP BRAFi (n = 12)–50.67, SEM 7.26 vs. MCR:SATB2 DMSO (n = 17) +11.06, SEM 1.84 vs. MCR:SATB2 BRAFi (n = 16) +22.06, SEM 2.81. Unpaired two-tailed t-test p values of pairwise comparisons are shown.

very large tumors (*Figure 1D–G*, *Figure 1—figure supplement 2D–E*). This has not been previously observed with several other accelerators or drug resistance drivers that we and others have identified using the MiniCoopR model (*Ablain et al., 2018*; *Ceol et al., 2011*; *Dang et al., 2016*; *Fazio et al., 2020*; *Iwanaga et al., 2020*; *Kaufman et al., 2016*; *Venkatesan et al., 2018*).

This phenotype could be explained by a proliferation effect resulting in multiple parallel tumor initiation events in both dermal and internal melanocytes (*Kapp et al., 2018*), or by an early invasive migration and metastasis phenotype. We did not observe proliferation differences between MCR:SATB2 and MCR:EGFP (*Figure 1—figure supplement 4A*), nor in a panel of three human melanoma cell lines upon inducible overexpression of SATB2 (*Figure 1—figure supplement 4B–C*). On the other hand SATB2 overexpression in cultured zebrafish tumors (*Figure 2A* and *Figure 2—figure supplement 1A–D*) and human melanoma cell lines (*Figure 2A*, *Figure 2—figure supplement 1E*, and *Video 2*) resulted in increased invasion in vitro, and MCR:SATB2 zebrafish primary tumor allografts in irradiated optically clear *casper* recipients had a much higher distant and internal metastasis rate compared to MCR:EGFP controls (*Figure 2F–J*, *Figure 2—figure supplement 2D*, and *Videos 3–4*), suggesting the latter hypothesis to be more likely at the basis of our observations in tumor-bearing MCR:SATB2 transgenic fish.

Our chromatin and transcriptional analysis via ChIP-seq, and RNA-seq followed by qPCR validation on MCR:SATB2 vs. MCR:EGFP control tumors showed SATB2 to directly bind and transcriptionally induce several genes related to neural crest development (*snai2* and *chd7*) (*Bajpai et al., 2010*; *Betancur et al., 2010*; *Mayor and Theveneau, 2013*; *Schulz et al., 2014*), EMT (*vim* and *fn1*) (*Betancur et al., 2010*), and invadopodia formation (*pdgfab* and *rhobtb2*) (*Maheswaran and Haber, 2015*; *Mayor and Theveneau, 2013*; *Murphy and Courtneidge, 2011*; *Nieto et al., 2016*; *Paz et al., 2014*). Functionally, an EMT-like phenotype and ECM remodeling in MCR:SATB2 vs. MCR:EGFP tumors was also apparent at the protein level (*Figure 3E*), and SATB2 overexpression induced the formation of functional invadopodia with ECM degrading capacity in zebrafish tumors, primary cell lines, and human melanoma cell lines alike (*Figure 2A–E*, *Figure 2—figure supplement 1B–E*, and *Video 2*). This neural crest mesenchyme-like program outlined above is consistent with SATB2's known roles in the development of exo-mesenchymal derivatives of the CNC (*Figure 3—figure supplement 1E–F*; *Ahn et al., 2010*; *Hassan et al., 2010*; *Kikuiri et al., 2018*; *Nanni et al., 2019*; *Rainger et al., 2014*; *Sheehan-Rooney et al., 2010*; *Sheehan-Rooney et al., 2013*; *Zarate et al., 2019*; *Zarate and Fish, 2017*; *Zhou et al., 2016*), neuronal development and axon guidance (*Gyorgy et al., 2008*; *McKenna et al., 2015*; *Shinmyo and Kawasaki, 2017*), and regulation of EMT/invadopodia in other solid tumors (*Gan et al., 2017*; *Mansour et al., 2015*; *Naik and Galande, 2019*; *Wu et al., 2016*; *Xu et al., 2017*; *Yu et al., 2017a*). EMT and invadopodia formation are, in fact, inherent to migration of the neural crest itself, since neural crest cells are specified in proximity of the neural plate, undergo EMT and rely upon podosomes, termed invadopodia in cancer cells, to remodel the ECM and migrate through the body (*Bailey et al., 2012*; *Betancur et al., 2010*; *Mayor and Theveneau, 2013*; *Murphy and Courtneidge, 2011*; *Murphy et al., 2011*). Reactivation of neural crest regulator *snai2* has been shown to increase the transformation and migration potential of melanoma (*Gupta et al., 2005*; *Shirley et al., 2012*). Overexpression of human SNAI2 in the MiniCoopR model similarly resulted in faster tumor development compared to EGFP controls (Log-rank (Mantel-Cox) test p<0.0001), although not to the same extent as SATB2 (MCR:SNAI2 median onset 18 weeks, MCR:SATB2 median onset 12 weeks, MCR:EGFP median onset ~22 weeks in both experiments; (*Figure 1C* and *Figure 3F–G*)) and similar invasive histological features compared to MCR:SATB2 tumors (*Figure 3H*). Taken together, these data suggest that downstream SATB2 target *snai2* is partially responsible for the SATB2 phenotype, but that other genes in the program contribute to the full phenotype spectrum. PDGF ligands and their receptor PDGFRB are of particular interest as additional downstream effectors, and further studies should be performed given their known role in EMT (*Ekpe-Adewuyi et al., 2016*), invadopodia formation (*Charbonneau et al., 2016*; *Murphy and Courtneidge, 2011*; *Quintavalle et al., 2010*), and acquired resistance to BRAF inhibition (*Nazarian et al., 2010*; *Rebecca et al., 2014*).

To further assess the conservation and relevance of our findings to human melanoma, we conducted qPCR analysis on a panel of iSATB2 human melanoma cell lines and primary cultured melanocytes (HEMa-LP). qPCR of the human orthologs of *snai2*, *pdgfab*, and *sema3f* overall showed a similar induction as zebrafish tumors, with some variation across cell lines (*Figures 3D* and *4A*). Even though our data seem to suggest an overall program and phenotype conservation, it is likely that SATB2 transcriptional effect is to a degree context dependent and might be affected by the baseline expression levels of specific genes across cell lines (e.g. MITF baseline status). For an unbiased global transcriptional comparison, we conducted RNA-seq on human melanoma NRAS-driven cell line SKMEL2 iSATB2, since it had the strongest invadopodia phenotype, and identified transcriptional differences induced by SATB2 overexpression (with/without Dox induction for 48 hr). Significant differentially regulated genes from SKMEL2 iSATB2 were compared with zebrafish primary tumor RNA-seq (MCR:SATB2 vs. MCR:EGFP) using both Gene Set Enrichment Analysis (GSEA) and Ingenuity Pathway Analysis (IPA). Despite (1) the ortholog prediction constraints between human and zebrafish datasets that may have resulted in some expected loss of information, (2) the variation due to species genetic and functional conservation differences, and (3) the in vivo vs. in vitro conditions, we see strong conservation across the two models in the top altered pathways (*Figure 4B*). Overall, our study shows a robust transcriptional and phenotypic conservation of SATB2 overexpression effects in melanoma across human and zebrafish (*Figure 2A–E*, *Figure 2—figure supplement 1B–E*, *Figure 3D*, *Figure 4B–C*, *Figure 4—figure supplement 1A–B*, and *Figure 1—figure supplement 4A,C*). The SATB2-driven program we describe here (*Figure 4C–D*) significantly overlaps with the recently reported MITF^low/AXL ^high state (*Tirosh et al., 2016*), and the neural crest-like MITF^low/NGFR1^high/AQP1^high state (*Rambow et al., 2018*), which are known to be less proliferative and resistant to targeted inhibitors of the MAPK pathway. It should be further noted that a similar positive correlation between the MITF^low/AXL ^high state, PDGFRB high expression, and acquired resistance to MAPK pathway targeted inhibitors in both BRAF (e.g. SKMEL28) and NRAS (e.g. SKMEL2) mutated melanoma cell line has been previously observed (*Müller et al., 2014*). Indeed, SATB2 overexpression in SKMEL2 (*Figure 4C*, *Figure 4—figure supplement 1A*) induced upregulation of AXL, AQP1, RXRG, NGFR, PDGFRB, and SRC, which have all individually been reported to drive resistance to MAPK pathway inhibitors (*Boshuizen et al., 2020*; *Nazarian et al., 2010*; *Fallahi-Sichani et al., 2017*). Interestingly, *SATB2* has been reported as a primary hit in an unbiased overexpression screen for MAPK pathway inhibitor resistance drivers. Specifically, it was showed to induce resistance across four human melanoma cell lines to BRAF inhibitor PLX4720, which is the progenitor compound of FDA-approved BRAF inhibitor Vemurafenib (PLX4032) (*Johannessen et al., 2013*). We sought out to functionally validate SATB2 as a resistance driver by conducting in vivo limiting dilution transplants and drug treatments with Vemurafenib using established assays in zebrafish allografts (*Dang et al., 2016*; *Heilmann et al., 2015*) and showed MCR:SATB2 tumors to have increased tumor propagating potential (*Figure 5A–C*), and primary resistance to Vemurafenib treatment in vivo (*Figure 5D–F*). While genetic conservation and/or pharmacokinetic differences can limit the study of drug resistance in zebrafish, the cross-species phenotype conservation between our in vivo zebrafish data, together with the in vitro evidence in human melanoma cell lines from *Johannessen et al., 2013* validate *SATB2* as driver of resistance to BRAF inhibition.

Analysis of publicly available genomic datasets showed *SATB2* to not be recurrently altered in the TCGA dataset (*ICGC/TCGA Pan-Cancer Analysis of Whole Genomes Consortium, 2020*), but its amplifications is observed in ~4–8% of patients in three independent datasets of metastatic melanoma (*Figure 2—figure supplement 3C*; *Hugo et al., 2015*; *Snyder et al., 2014*; *Van Allen et al., 2014*), with a similar fraction of SATB2 high-expressing patients having a higher risk of metastasis-related poor outcome in two additional datasets (*Figure 2—figure supplement 3D*).

In summary, our work identifies SATB2 as a driver of invasion and resistance to Vemurafenib treatment in melanoma. Yet, further studies focused on the analysis of patient samples will be needed to assess the prevalence, and genetic/transcriptional context of SATB2 expression changes in targeted therapy and immunotherapy clinical settings. Furthermore, we show in an autochthonous transgenic tumor model that SATB2 transcriptional rewiring of melanoma towards a state similar to the less proliferative neural crest-like state described by *Rambow et al., 2018* can drive accelerated tumor development and invasion. Our work reinforces the idea that melanoma phenotype switching and transcriptional plasticity might dynamically select a dominant state depending on environmental

constraints on proliferation during different stages of the tumor's natural history (*Ahmed and Haass, 2018*; *Boumahdi and de Sauvage, 2020*; *Li et al., 2015*; *Marine et al., 2020*).

## Materials and methods

### Zebrafish melanoma model and MiniCoopR system

Experiments were performed as outlined by *Ceol et al., 2011*. MiniCoopR (MCR) expression constructs were created by MultiSite Gateway recombination (Invitrogen) using full-length human open-reading frames. Briefly, pools of five similarly sized MiniCoopR constructs (5 pg each), or single factors (25 pg) including MCR alone, MCR:EGFP (*Ceol et al., 2011*), MCR:SATB2 (IOH46688, Invitrogen), MCR:SNAI2 [pCMV-SPORT6-SNAI2 HsCD00327569],and MCR:SETDB1 (*Ceol et al., 2011*), were microinjected together with 25 pg of Tol2 transposase mRNA into one-cell Tg(*BRAF^{V600E}*); *p53^{-/-}*; *mitf^{-/-}* zebrafish embryos. Embryos were scored for melanocyte rescue at 48–72 hr post-fertilization, and equal numbers were raised to adulthood (15–20 zebrafish per tank), and scored weekly (from 8 to 12 weeks post-fertilization) or bi-weekly (>12 weeks post-fertilization) for the emergence of melanoma. Macroscopically visible pigmented raised lesions were scored as melanoma as previously described (*Ceol et al., 2011*). Prior work establishing the melanoma transgenic model showed raised lesions to be histologically and functionally malignant tumors, while benign nevi to be hyper pigmented but to remain flat and not be able to propagate tumors (*Ceol et al., 2011*; *Kaufman et al., 2016*; *McConnell et al., 2019*; *Patton et al., 2005*). The minimum cohort size per injection was 40 animals as previously defined (*Ceol et al., 2011*). We pre-established criteria to exclude/censor animals if they died without having been examined for tumor formation. No randomization or blinding was used. Kaplan-Meier survival curves were generated in GraphPad Prism, and statistical difference was determined by a log-rank (Mantel-Cox) test. For subsequent expression, binding and histopathological analyses, large tumors were isolated from MCR:SATB2 (9–18 weeks post-fertilization), MCR/MCR:EGFP (14–28 weeks post-fertilization, due to later tumor onset and slower progression compared to MCR:SATB2) MCR:SNAI2(14–20 weeks post-fertilization). Zebrafish were maintained under IACUC-approved conditions (Boston Children's Hospital Institutional Animal Care and Use Committee protocol # 20-10-4253R).

### CRISPR/Cas9 inactivation of *satb2*

To specifically inactivate *satb2* in melanocytes, we engineered the MiniCoopR vector to express Cas9 under the control of the melanocyte-specific *mitfa* promoter, and a gRNA efficiently mutating *satb2* off a *U6* promoter (*Ablain et al., 2015*). *Cas9* mRNA was produced by in vitro transcription from a pCS2 *Cas9* vector (*Jao et al., 2013*) using mMESSAGE mMACHINE SP6 kit (Invitrogen). The *satb2* gRNA 5'-GGATGGGCAGGGGGTTCCAG-3' was generated following established methods (*Gagnon et al., 2014*). To validate targeting, 600 pg of *Cas9* mRNA and 25 pg of gRNA were injected into embryos of the AB strain, and the T7E1 assay was performed as reported (*Kim et al., 2009*), using the following primers: 5'-CCTACCTCAATCCACTCTTT-3' and 5'-GCTGCACCAA-GAAACTACAA-3'. Verified gRNAs against *satb2,* and a *p53* control (*Ablain et al., 2015*) were injected into one-cell stage Tg(*mitfa:BRAF^{V600E}*); *p53^{-/-}*; *mitfa^{-/-}* embryos, and tumor formation was monitored.

### Morpholino knockdown of *satb2*

To address whether SATB2 is necessary for neural crest and melanocyte development, 4 pg of a splicing morpholino against *satb2* (GCAGTGTTGAACTCACCATGAGCCT, *Ahn et al., 2010*) was injected into one-cell stage TG(*sox10:mCherry*) embryos. At 3 dpf, injected and uninjected control embryos were scored for melanocyte and cranio-facial abnormalities using light and fluorescence microscopy. The experiment was repeated three times, with an average clutch size of 40 embryos per experiment.

### Zebrafish primary tumor and cell line transplants

Pigmented primary melanomas were excised from euthanized MCR/MCR:EGFP and MCR:SATB2 zebrafish, and dissociated in 50% Ham's-12/DMEM medium containing 0.075 mg/mL liberase (Promega) for 30 min at RT, with periodical manual disaggregation using a razor blade. The dissociation

medium was inactivated by the addition of 3 × 5 ml 50% Ham's-12/DMEM with 15% heat-inactivated Fetal Bovine Serum. To obtain a single-cell suspension, cells were passaged through a 40 μm filter into a 50 mL falcon tube, and pelleted by centrifugation at 500 x g for 5 min. Cell numbers were determined, and a cell suspension of 100,000 cells/μl in PBS was made, and kept on ice. Casper recipients received a split-dose irradiation of 15 Gy over 2 consecutive days prior to transplantation (*Heilmann et al., 2015*). During transplantation, irradiated Casper recipients were anesthetized in 0.4% MS222, mounted in a moist sponge. Using a Hamilton syringe, 300,000 pigmented primary tumor cells, or 250,000 MCR:EGFP (zmel1) or MCR:SATB2 (45–3) cells were injected into a confined interstitial subcutaneous space that runs along the dorsum of the zebrafish. Five to seven primary tumors were used per cohort, with each individual tumor being transplanted into 6–12 *casper* recipients. Metastatic progression, as defined by the formation of distant metastasis that have spread passed the anatomical midline, was monitored weekly or at the experimental end point at 3–3.5 weeks post-transplant, and photographically recorded using a Nikon D3100 DSLR camera. Limiting dilution experiments were performed in a similar fashion by using multiple dilutions from each individual donor and transplanting cohorts of 10 recipients per each experiment arm. Overall 4 MCR : EGFP donors and 5 MCR : SATB2 donors were transplanted in the course of three independent experiments. Recipients were monitored weekly for engraftment or at the experimental end point at 3 weeks post-transplant, and photographically recorded using a Nikon D3100 DSLR camera. The percentage of engraftment and n was used as input to estimate the fraction of tumor propagating cells using Extreme limiting dilution analysis ELDA (*Hu and Smyth, 2009*) Statistical difference across groups was calculated with a Mann-Whitney test using GraphPad Prism. No blinding was performed, and the group size was not predetermined statistically. Statistical significance for all above experiments was determined by an unpaired two-tailed *t*-test unless otherwise specified, using GraphPad Prism. Drug treatment with Vemurafenib (100 mg/kg dissolved in DMSO) or with equal amount of DMSO dissolved in water was performed via daily oral gavage for 14 days starting on day 10 post-transplant on irradiated *casper* recipients allografted in the peritoneum with 500,000 pigmented primary tumor cells. Recipients were monitored at day 10 post-transplant before starting treatment, and at the experimental end point at day 24 post-transplant (14 of treatment), and photographically recorded using a Nikon D3100 DSLR camera. Tumor response was assessed as previously described Tumor area was measured at day 10 and day 24 using a traceable digital caliper (Fisher, 14-648-17). The pigmented tumor area was calculated by the longest measured length and width of the tumor. The drug response was quantified via the change from baseline tumor area using the RECIST (Response Efficacy Criteria in Solid Tumors). The response rate for experimental cohorts was depicted via waterfall plots, and *t*-test statistics were applied for significance as previously described (*Dang et al., 2016*).

## Whole-mount tumor immunohistochemistry

Zebrafish were euthanized on ice, melanomas were surgically removed, and fixed in either 4% paraformaldehyde (PFA) or DENTS (80% methanol, 20% DMSO) overnight at 4°C. Tumors were washed at least 3 times with PBS, and mounted in 4% agarose in PBS for Vibratome sectioning. One hundred to 150 μM sections were cut using the Microm HM650V Vibratome. Sections were next processed for immunohistochemistry. PFA fixed samples were permeabilized by digestion with 15 μg/ml Proteinase K in PBS-0.1% Tween for 30 min at 37°C, or 0.4% Triton in PBS for 10 min at room temperature. After washing three times with PBS-0.1% Tween for 10 min, samples were blocked (10% lamb serum, 1% DMSO in PBS-0.1% Tween) for at least 1 hr, and incubated with primary antibodies overnight at 4°C. Antibodies used were: anti-E Cadherin (1:100, DENTs fixation; ab11512, Abcam), Anti-Vimentin (1:200; sc-6260, Santa-Cruz), anti-Collagen I (1:250; ab23730, Abcam), anti-Fibronectin 1 (1:250; F3648, Sigma Aldrich), and anti-Cortactin (p80/85), clone 4F11 (1:100; 05–180, EMD Millipore). After extensive washes with PBS-0.1% Tween, and blocking for 1 additional hour, samples were incubated in appropriate secondary Alexa-Fluor antibodies (1:500 Invitrogen) overnight at 4°C. Nuclei were counterstained with 1:1000 DAPI, and sections were mounted on a microscope slides in Slow Fade Gold (Invitrogen). Confocal images were collected on a Nikon C2si Laser Scanning Confocal using 40x water or 63x oil immersion objectives.

## Tumor histology and proliferation index

Zebrafish were fixed in 4% paraformaldehyde, decalcified, paraffin-embedded, sectioned (7 µm) and Hematoxylin and Eosin (H and E) stained by the DFCI/HCC Research and BWH pathology cores, using standard procedures. Transmitted light images were collected on a Nikon C2si Laser Scanning Confocal using a Hamamatsu camera. To determine proliferation rates, slides were incubated with Anti-Histone H3 (phospho S10; 1:250; ab5176, Abcam), and colorimetrically stained with DAB following standard procedures. The average number of PH3-positive cells per tumor was determined by counting five high-power field images (×40–×60) per tumor. An unpaired tailed Student's T-test was used to determine significance.

## Western blot

Adherent cells were scraped, and zebrafish tumors were mechanically homogenized in RIPA lysis buffer containing 1:100 protease inhibitors (P8340, Sigma-Aldrich) and 20 µM N-ethylmaleimide (Sigma-Aldrich). Lysates were incubated for 20 min on ice, and spun down for 10 min at 14,000 RPM at 4°C. Samples were denatured by adding Laemmli sample buffer (BioRad) with 5% β-mercaptoethanol (Sigma-Aldrich), and boiled at 95°C for 5 min prior to loading. Protein concentrations were determined using the DC protein assay (BioRad). Proteins (20 µg) were separated on a 4–20% mini-PROTEAN TGX (BioRad) precast gel, and transferred onto a nitrocellulose membrane using the iBlotting system (Invitrogen). Primary antibodies used were: Anti-SATB2 (zebrafish tumors, 1 ug/ml; Ab51502, Abcam), Anti-SATB2 (human cells, 1:200; HPA029543, Sigma Aldrich) and Anti-Beta Actin (1 µg/ml; A2228, Sigma-Aldrich) as a loading control. Additional antibodies used against SATB1 and SATB2 are listed in *Figure 2—figure supplement 3B* and *Figure 3—figure supplement 1A*. Protein bands were detected by rabbit anti-mouse HRP (1:20,000, Pierce) or swine anti-rabbit HRP (1:20,000, Pierce).

## RNA extraction, quantitative RT-PCR analysis, and RNA-seq sample preparation

Zebrafish tumors were isolated and mechanically homogenized in 350 µl RTL buffer (Qiagen) containing β-mercaptoethanol, on ice for 20 s. Adherent cells (zebrafish and human cell culture) were washed twice with ice cold PBS on ice, and cells were scraped in RTL buffer containing β-mercaptoethanol (Sigma-Aldrich). Tumor and cell lysates were next transferred onto a QiaShredder column (Qiagen). RNA isolation was performed using the RNA micro plus kit (Qiagen), according to the manufacturers instruction. RNA quality was determined using a Nanodrop. For RNA-seq on primary zebrafish melanomas, additional quality control of the total RNA was performed on and Fragment Analyzer. Total RNA was depleted of ribosomal RNA with the RiboZero gold kit (Epicentre), and enriched mRNA was applied to library preparation according to manufacturer's protocol (NEBNext Ultra). After repeated quality control for an average DNA input size of 300 base pairs (bp), samples were sequenced on a HiSeq Illumina sequencer with 2 × 100 bp paired-end reads. For qPCR, cDNA was synthesized with the SuperScript III Kit (Life Technologies) using a 1:1 mixture of random Hexamers and OligoDT. Quantitative PCR was performed on a BioRad iQ5 real-time PCR machine using the Ssofast EvaGreen Supermix (BioRad). The ΔCt or ΔΔCt methods were used for relative quantification. The average of two independent experiments is shown for SKMEL2 and a single experiment is shown for primary human melanocytes. qPCR primers were designed using GETprime (*Gubelmann et al., 2011*) or qPrimerDepot (https://primerdepot.nci.nih.gov). For primer sequences see *Supplementary file 5*.

## Zebrafish primary melanoma cell culture

Zebrafish primary melanoma cell lines were generated as described (*Heilmann et al., 2015*). MCR alone (CK5), MCR:EGFP (zmel1), and MCR:SATB2 (45–3 and 63–4) cell lines were cultured in DMEM medium (Life Technologies) supplemented with 10% heat-inactivated FBS (Atlanta Biologicals), 1X GlutaMAX (Life Technologies) and 1% Penicillin-Streptomycin (Life Technologies), at 28°C, 5% CO₂. Zebrafish melanoma lines were authenticated by qPCR and western for human SATB2 or EGFP transgene expression, and periodically checked for mycoplasma using the Universal Mycoplasma Detection Kit (ATCC).

## Human melanoma and primary melanocytes cell culture

Human melanoma cell lines A375, SKMEL2, and SKMEL28 were obtained from the ATCC. Their identity was verified by ATCC, and they were further were regularly authenticated based on their distinct morphology. These human melanoma cell lines are not reported in the database of commonly misidentified cell lines maintained by ICLAC. They were grown in DMEM medium (Life Technologies) supplemented with 10% heat-inactivated FBS (Atlanta Biologicals), 1X GlutaMAX (Life Technologies) and 1% Penicillin-Streptomycin (Life Technologies), and grown at 37°C, 5% $CO_2$, and periodically checked for mycoplasma using the Universal Mycoplasma Detection Kit (ATCC). Primary Human Epidermal Melanocytes from an adult lightly pigmented donor, (HEMa-LP) were obtained from Thermo-Fisher Scientific and cultured according to manufacturer instructions in Medium 254 supplemented with Human Melanocyte Growth Supplement-2.

## Transduction of human melanoma cell lines and primary melanocytes

The full-length human SATB2 CDS (IOH46688, Invitrogen) was cloned into pINDUCER20 (*Meerbrey et al., 2011*) and pLenti CMV Blast DEST (706-1) (Addgene plasmid #17451 a gift from Eric Campeau and Paul Kaufman) via Gateway recombination using LR clonase II (Invitrogen) according to the manufacturers instructions. Lentiviral particles were produced by co-transfection of 293 T cells with sequence verified pINDUCER20-SATB2, pLentiCMV_SATB2_Blast, or pTK93_Lifeact (Addgene plasmid #46357 was a gift from Iain Cheeseman), and packaging plasmids pMD2.G (Addgene plasmid #12259) and psPAX2 (Addgene plasmid #12260, both gifts from Didier Trono), using FuGENE HD (Promega). Viral particles were harvested 48 and 72 hr post-transfection, concentrated by overnight PEG precipitation (*Kutner et al., 2009*), resuspended in PBS, and stored at −80°C. Human melanoma cell lines and Hema-LP were overlaid with viral particles diluted in DMEM/1x Glutamax with 10% TET System Approved FBS (Clontech) supplemented with 5 µg/ml polybrene (Sigma-Aldrich), for 24 hr at 37°C. 48 hr post-transduction, infected cells were selected with 500 µg/ml G418 (Gibco) for 7 days (pINDUCER20-SATB2), or with 2 µg/ml puromycin (Gibco) for 3 days (pTK93_Lifeact), or with 10 µg/ml blasticidin (Gibco) for 3 days (pLentiCMV_SATB2_Blast) replacing selection medium every 48 hr.

## Matrix degradation assay

Oregon green 488-conjugated gelatin covered 12 mm coverslips, or glass bottom 6-well plates were coated as described (*Martin et al., 2012*). Human melanoma cell lines were induced with doxycycline for 48 hr prior to plating. Human and zebrafish cells were plated at a density of 30,000 cells/well in a 12-well plate, and grown on the fluorescent gelatin for 23–25 hr at 37°C (human cells) or 28°C (zebrafish cells). Cells were fixed in 4% PFA for 10 min at room temperature, washed three times with PBS, permeabilized with 0.4% Triton-PBS for 4 min, followed by 30 min blocking (10% lab serum, 1% DMSO, 0.1% Tween in PBS). Coverslips were next stained with Alexa 650- or 568-conjugated Phalloidin (1:50) in blocking buffer, overnight at 4°C. Coverslips were washed extensively and nuclei were stained with DAPI, 10 min in PBS/0.1%Tween at RT. Coverslips were inversely mounted in Slow Fade Gold (Invitrogen) on a slide, and imaged on a Nikon C2si Laser Scanning Confocal, using ElementsX software. Experiments were seeded in triplicate, in at least three independent experiments. Four to six high-power (40-60x) areas per coverslip were imaged to determine the fraction of cells with degraded gelatin. An unpaired two-tailed *t*-test was used to compare significance between groups.

## Time-lapse video of fluorescent gelatin degradation after SATB2 induction

Induced human melanoma cell lines were seeded onto an Oregon green 488-conjungated gelatin coated glass bottom 6-well plate, and allowed to adhere for 3 hr at, prior to imagining. Time-lapse movies were recorded on a Nikon Eclipse Ti Spinning Disk Confocal with a ×10 objective (70 µm, 4 µm step size), for 16 hr in 20 min intervals at 37°C, 5% $CO_2$. Images were processed using Photoshop, ImageJ, or Imaris.

## Proliferation assays

Proliferation rates were determined using the CellTiter Glo (Promega) luminescent cell viability assay, according to the manufacturer instructions. Cells were seeded at an initial density of 5000 or 10,000 cells/well (98 well plate), in triplicate or quadruplicate wells, and experiments were repeated at least three independent times. An unpaired two-tailed $t$-test was used to compare significance between groups.

## ChIP-seq tumor sample preparation and sequencing

ChIP-seq was performed as previously described (*Lee et al., 2006*). Zebrafish were euthanized on ice and melanomas were excised, finely minced using a scalpel blade in 5 ml cold PBS in a petri dish, transferred to a 50 ml Falcon tube, and the petri dish was rinsed once with 5 ml PBS. Tumor samples were cross-linked in 1:10 11% formaldehyde solution for 10 min at RT, with occasional gentle mixing. Formaldehyde was quenched by addition of 1:20 2.5M glycine. Next, the tumor suspension was mechanically homogenized on ice at the lowest setting for 30 s or until no large tumor chunks are visible and passed through a 100 µm filter into a new 50 ml Falcon tube. Cells were spun down at 2500 RPM for 15 min, washed twice with PBS, and pellets were flash frozen and stored at −80°C, or subjected to chromatin immunoprecipitation with anti-SATB2 (sc-81376, Santa Cruz), anti-Histone H3 acetyl K27 (ab4729, Abcam) or anti-Histone H3 tri methyl K9 (ab8898, Abcam) antibodies. Ten µl of input DNA and the entire volume of ChIP DNA samples were prepared for sequencing. Libraries were prepared using the NEBNext Multiplex Oligos Kit (18-cycle PCR enrichment step), mixed in equal quantities (2–10 nM), and run on an Illumina Hi-Seq2000.

## ChIP-sequencing bioinformatics

All ChIP-Seq datasets were aligned to build version danRer7 of the zebrafish genome using Bowtie2 (version 2.2.1) (*Langmead and Salzberg, 2012*) with the following parameters: --end-to-end, -N0, -L20. We used the MACS2 version 2.1.0 (*Zhang et al., 2008*) peak finding algorithm to identify regions of ChIP-Seq peaks, with a $q$-value threshold of enrichment of 0.05 for all datasets (*Langmead and Salzberg, 2012*). Super enhancers were called using the ROSE package with the following parameters: -STITCHING_DISTANCE = 12.5, -TSS_EXCLUSION_ZONE_SIZE = 0. The H3K27Ac ChIP-Seq called peaks are used as enhancer gff input (*Lovén et al., 2013*; *Whyte et al., 2013*). Genome track images were generated using the UCSC browser. SATB2-bound loci were determined by overlapping all regions 3 kb upstream of the transcriptional start site (TSS), gene body (GB) and 3 kb downstream of the transcriptional end site (TES) of all transcripts in the danRer7 assembly, with all significantly bound SATB2-peaks ($p < 10^{-7}$). SATB2-bound peaks were compared to SATB2 enhancers peaks to determine the SATB2-bound enhancers. GREAT (*Hiller et al., 2013*) version 2.0.2 was used for GO-term enrichment analysis of SATB2-associated loci and SATB2-bound enhancers, using the 'Basal plus extension' genomic region association (proximal 10 kb upstream of TSS, 5 kb downstream of TSS, distal up to 100 kb) to associate genomic regions with nearby genes. The list of SATB2-bound genes was converted to human orthologs using DIOPT - DRSC Integrative Ortholog Prediction Tool (*Hu et al., 2011*) returning only the best matching ortholog. Data sets are deposited to the GEO Gene Expression Omnibus, accession number GSE77923 (http://www.ncbi.nlm.nih.gov/geo/query/acc.cgi?acc=GSE77923).

## RNA-sequencing bioinformatics

RNA-seq was performed on three biological replicates of primary zebrafish melanoma tumors that were excised from MCR:EGFP (control) and MCR:SATB2 overexpressing Tg(*BRAF*^V600E^);*p53*^-/-^; *mitf*^-/-^ zebrafish. Zebrafish were euthanized on ice, melanomas were isolated and mechanically homogenized in RTL buffer (Qiagen) containing β-mercaptoethanol. Tumor lysates were transferred onto a QiaShredder column (Qiagen), and RNA isolation was performed using the RNA Micro Plus kit (Qiagen) according to the manufacturers instruction. RNA quality was determined using a Nanodrop and Fragment Analyzer (Advanced Analytical). Total RNA was depleted of ribosomal RNA with the Ribo-Zero Gold kit (Epicentre). Ribosome-depleted RNA was used to create multiplexed RNA-seq libraries (NEBNext Ultra, NEB) according to manufacturer's instructions. Samples with an average DNA input size of 300 base pairs were sequenced on an Illumina Hi-Seq2000 sequencer with $2 \times 100$ bp paired-end reads. Quality control of RNA-Seq datasets was performed by FastQC29 and

Cutadapt30 to remove adaptor sequences and low quality regions. The high-quality reads were aligned to UCSC build danRer7 of zebrafish genome using Tophat31 2.0.11 without novel splicing form calls. Transcript abundance and differential expression were calculated with Cufflinks32 2.2.1. FPKM values were used to normalize and quantify each transcript; the resulting list of differential expressed genes are filtered by log(2) fold change >1.5 and a p-value<0.05. Human orthologs were predicted using DIOPT33. Data sets are deposited to the GEO Gene Expression Omnibus accession number GSE77923. RNA-seq was performed on three biological replicates of SKMEL2 transduced with pINDUCER20-SATB2 after 48 hr of induction with 2 µg/mL of doxycycline as outlined above, except for the mechanical homogenization. Pathway analyses were conducted using GSEA (v4.0) (*Subramanian et al., 2005*) and IPA (Qiagen). Pathway and gene list overlaps were plotted using BioVenn (*Hulsen et al., 2008*).

## Statistics

Comparison of Kaplan-Meier survival curves was performed by a log-rank (Mantel-Cox) test. Statistical difference in qPCR expression analysis between large groups of MCR:EGFP and MCR:SATB2 primary tumors, and difference in estimated fraction of tumor propagating cells between MCR:EGFP and MCR:SATB2 donors were determined by an unpaired two-tailed Mann-Whitney test. The rest of the statistics were performed with an unpaired two-tailed t-test. An F-test was used to determine similar variation between compared groups. Graphs show the median with s.e.m. No statistical methods were used to predetermine sample size. Experiments to quantify proliferation by PH3 immunohistochemistry, and cells with degraded 488-conjugated gelatin were scored blindly. Irradiated Casper zebrafish were randomized between transplantation groups. All statistical analyses were performed with GraphPad Prism. NS, not significant, p>0.05; *p≤0.05; **p≤0.01; ***p≤0.001; ****p≤0.0001.

## Acknowledgements

We thank Serine Avagyan and Alicia McConnell for helpful discussion, Elliott Hagedorn for helpful discussion and time-lapse imaging support, Cristine Lian for histopathology discussion, and Rachel Fogely for ChIP-seq guidance. MF was supported by Boehringer Ingelheim Fonds. EvR was supported by a Netherlands Organization for Scientific Research (NWO) Rubicon fellowship, and a Dutch Cancer Foundation (KWF) fellowship for Fundamental Cancer Research. LIZ is supported by: R01 CA103846, MRA (Zon, Garraway), The Starr Cancer Consortium and the Ellison Foundation.

## Additional information

### Competing interests

Richard M White: Senior editor, *eLife*. Leonard I Zon: LIZ is a founder and stockholder of Fate Therapeutics Inc, Scholar Rock Inc, Camp4 Therapeutics Inc, Amagma Therapeutics Inc, and a scientific advisor for Stemgent. The other authors declare that no competing interests exist.

### Funding

| Funder | Grant reference number | Author |
| --- | --- | --- |
| Boehringer Ingelheim Fonds | | Maurizio Fazio |
| Netherlands Organisation for Scientific Research | Rubico Fellowship | Ellen van Rooijen |
| Dutch Cancer Society | | Ellen van Rooijen |
| National Cancer Institute | R01 CA103846 | Leonard I Zon |
| Melanoma Research Alliance | | Leonard I Zon |
| Starr Foundation | | Richard M White Leonard I Zon |
| Ellison Medical Foundation | | Leonard I Zon |

The funders had no role in study design, data collection and interpretation, or the decision to submit the work for publication.

## Author contributions
Maurizio Fazio, Conceptualization, Resources, Data curation, Formal analysis, Validation, Investigation, Visualization, Methodology, Writing - original draft, Writing - review and editing; Ellen van Rooijen, Conceptualization, Resources, Data curation, Software, Formal analysis, Validation, Investigation, Visualization, Methodology, Writing - original draft, Writing - review and editing; Michelle Dang, Glenn van de Hoek, Andrew Thomas, Jonathan Michael, Tania Fabo, Rodsy Modhurima, Investigation; Julien Ablain, Resources, Formal analysis, Investigation, Methodology; Jeffrey K Mito, Resources, Software, Formal analysis, Investigation, Visualization; Song Yang, Resources, Data curation, Software, Investigation, Visualization, Methodology; Patrizia Pessina, Yi Zhou, Resources; Charles K Kaufman, Resources, Writing - review and editing; Richard M White, Resources, Methodology, Writing - review and editing; Leonard I Zon, Conceptualization, Formal analysis, Supervision, Funding acquisition, Investigation, Project administration, Writing - review and editing

## Author ORCIDs
Maurizio Fazio (iD) https://orcid.org/0000-0003-0083-6601
Tania Fabo (iD) http://orcid.org/0000-0002-8987-0672
Charles K Kaufman (iD) http://orcid.org/0000-0003-3122-1677
Richard M White (iD) http://orcid.org/0000-0001-9099-9169
Leonard I Zon (iD) https://orcid.org/0000-0003-0860-926X

## Ethics
Animal experimentation: Zebrafish were maintained under IACUC-approved conditions (Boston Children's Hospital Institutional Animal Care and Use Committee protocol # 20-10-4253R).

## Decision letter and Author response
Decision letter https://doi.org/10.7554/eLife.64370.sa1
Author response https://doi.org/10.7554/eLife.64370.sa2

## Additional files
### Supplementary files
• Supplementary file 1. Schematic of screened pools of epigenetic factors and source plasmid references.

• Supplementary file 2. Differential gene expression list of primary MCR:SATB2 vs MCR:EGFP tumors by RNA-seq (Sheet 1). Genes confirmed by qPCR on a large panel of primary MCR:SATB2 tumors versus MCR:EGFP tumors. Red is upregulated, green is downregulated. (Sheet 2). Best human ortholog prediction with DIOPT (Sheet 3). MCR tumors RNA-seq data matched with human orthologs (Sheet 4). Input for GSEA analysis on MCR tumors (Sheet 5). Input for IPA analysis on MCR tumors (Sheet 6). SATB2 bound genes based on SATB2 ChIP-seq in MCR:SATB2 tumor (Sheet 7). DIOPT human ortholog prediction of SATB2 bound targets (Sheet 8), and humanized SATB2 bound target list (Sheet 9) used for overlap in Biovenn with published TFAP2A active targets in *Figure 3D*. Overlap of significant genes with with ChIP-seq to determine SATB2-bound and SATB2-associated loci that are misexpressed. (Sheet 10).

• Supplementary file 3. SATB2-associated genes that are neural crest related (Sheet 1) SATB2-associated genes, determined by using GREAT genomic region association 'Basal plus extension' (proximal 10 kb upstream of TSS, 5 kb downstream of TSS, distal up to 100 kb). (Sheet 2) Neural-crest-associated genes. (Sheet 3) SATB2/neural-crest-associated genes. (Sheet 4) Human predicted orthologs (DIOPT) of SATB2-Bound genes (TSS+ gene body +/- 3 kb).

• Supplementary file 4. RNA-seq of SKMEL2 TetOn SATB2 48 hr +/- doxycycline. (Sheet 1) Genes with a log(2) fold change >1.0 and a p-value<0.05. (Sheet 2) Unfiltered RNA-seq expression data.

- Supplementary file 5. qRT-PCR primers.
- Transparent reporting form

## Data availability

Data sets are deposited to the GEO Gene Expression Omnibus, accession number GSE77923.

The following dataset was generated:

| Author(s) | Year | Dataset title | Dataset URL | Database and Identifier |
|---|---|---|---|---|
| Fazio M, van Rooijen E, Yang S, Zon LI, Fazio M | 2021 | SATB2 induces transcriptional programs in melanoma that lead to metastatic behavior | http://www.ncbi.nlm.nih.gov/geo/query/acc.cgi?acc=GSE77923 | NCBI Gene Expression Omnibus, GSE77923 |

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
