## [Decision Letter]

**Acceptance summary:**

Your manuscript using a powerful in vivo genetic system provides evidence that epigenetic regulators can be a driving force of malignant progression, specifically widespread metastatic disease. The data indicate relevance to human melanoma, both in disease progression and outcome and also in drug resistance.

**Decision letter after peer review:**

Thank you for submitting your article "SATB2 induction of a neural crest mesenchyme-like program drives invasion and drug resistance in melanoma" for consideration by *eLife*. Your article has been reviewed by three peer reviewers, including Grant McArthur as the Reviewing Editor and Reviewer #1, and the evaluation has been overseen by Marianne Bronner as the Senior Editor. The following individuals involved in review of your submission have agreed to reveal their identity: Patricia A Possik (Reviewer #2); Andrew G Cox (Reviewer #3).

The reviewers have discussed the reviews with one another and the Reviewing Editor has drafted this decision to help you prepare a revised submission.

Summary:

The manuscript by Fazio, Van Rooijen and colleagues reports an in vivo screen for acceleration of melanoma development in the well characterized BRAF/p53 Zebrafish model. The authors identify the transcriptional regulator SATB2 in the screen and show that SATB2 regulates a network of genes that induce a neural crest stem cell like phenotype with enhanced invasion and metastases and an EMT phenotype. They show amplification of SATB2 in a subset of human melanomas and also demonstrate resistance to the BRAF-inhibitor vemurafenib in vivo. Previous studies from other investigators have shown a role for SATB2 in EMT, invasion and metastasis in various cancer models and correlated expression with clinical outcomes in human samples. The current manuscript demonstrates that SATB2 can drive, through epigenetic mechanisms, metastasis in cooperation with oncogenic drivers in an in vivo model that allows understanding of SATB2, and hence epigenetic mechanisms, in disease progression.

Essential revisions:

It is not clear in the allograft experiments presented in Figure 2F-I and Figure 2—figure supplement 1 use cells derived from single EGFP or SATB2 derived tumors. Please revise to clarify this point and if these data are representative of multiple tumors?

One of the main claims of the study is that SATB2 is involved in drug resistance. However, this claim is supported (robustly) only by the findings in zebrafish. Human melanomas are highly mutated and genetically heterogeneous. It would be important to discuss plans to extend these data into the human setting recognizing possible limitations of studying resistance purely in Zebrafish.

Regarding the overexpression screen: Although the authors focus on gene pools that accelerate the onset of melanoma, several pools show the opposite effect. In fact, Figure 1B gives the impression that most of the significant pools decrease the onset of melanoma formation. According to the figure legends, 6 pools were significant. The fact that the authors do not list which are the significant ones and do not discuss this in the text is curious. Is there a specific (technical) reason for not looking into genes that delay melanoma onset? Of course, there needs to be focus on one question and for this study, the authors chose to focus on genes with a positive effect on tumor progression, but it would be important to mention which other pools scored significantly and perhaps add some discussion on that. If other genes or pools were validated, it would be important to add that as well.

---

## [Author Response]

Essential revisions:It is not clear in the allograft experiments presented in Figure 2F-I and Figure 2—figure supplement 1 use cells derived from single EGFP or SATB2 derived tumors. Please revise to clarify this point and if these data are representative of multiple tumors?

The data shown is the combined result of multiple cohorts. Individual donor tumors (7 donors from MCR: SATB2 and 5 donors from MCR:EGFP) were transplanted into cohorts of 8-12 recipients. We have now further clarified this in the Results and figure legends.

Results:

“At the experimental end point at 3-3.5 weeks-post transplantation, 59.4%±2.3% (SEM; n=76 total recipients grafted from 7 individual donor tumors) of MCR:SATB2 transplants formed distant metastasis, compared to 21.8%±4.5% (SEM; n=37 total recipients grafted from 5 individual donor tumors) of EGFP-control transplants (*p*<0.0001)”

Updated Figure 2 legend:

“(F) Pooled recipient data at the experimental end point at 3.5 weeks post-transplantation, 59.4%±2.3% (SEM; n=76 total recipients, grafted from 7 individual donor tumors) of MCR:SATB2 transplants formed distant metastasis, compared to 21.8 %±4.5% (SEM; n=37 total recipients grafted from 5 individual donor tumors) of EGFP-control transplants (*p*<0.0001).”

Updated Figure 2—figure supplement 2 (former Supplementary Figure 5) legend:

“(D) Weekly prospective imaging of primary MCR:SATB2 individual tumor donor allograft in irradiated casper recipient. Arrows indicate distant metastasis from injection site. Related to Figure 2H.”

One of the main claims of the study is that SATB2 is involved in drug resistance. However, this claim is supported (robustly) only by the findings in zebrafish. Human melanomas are highly mutated and genetically heterogeneous. It would be important to discuss plans to extend these data into the human setting recognizing possible limitations of studying resistance purely in Zebrafish.

To address this point we have extended our discussion of the relation between our functional in vivo data on drug resistance in zebrafish and the Johannessen et al., 2013 study, which showed SATB2 overexpression as a primary hit from their resistance screen in vitro to induce resistance to BRAF inhibitor PLX4720 across 4 melanoma cell lines. We also highlighted the necessity of this cross-species validation of zebrafish data with human data since, as suggested by the reviewers, species conservation or pharmacokinetic differences between zebrafish and human might otherwise limit and/or confound conclusions on drug resistance studies.

Updated Discussion text:

“Interestingly, SATB2 has been reported as a primary hit in an unbiased overexpression screen for MAPK pathway inhibitor resistance drivers. [...] While genetic conservation and/or pharmacokinetic differences can limit the study of drug resistance in zebrafish, the cross-species phenotype conservation between our in vivo zebrafish data, together with the in vitro evidence in human melanoma cell lines from Johannssen and colleagues (Johannessen et al., 2013) validate SATB2 as driver of resistance to BRAF inhibition.”

“In summary, our work identifies SATB2 as a driver of invasion and resistance to Vemurafenib treatment in melanoma. Yet, further studies focused on the analysis of patient samples will be needed to assess the prevalence, and genetic/transcriptional context of SATB2 expression changes in targeted therapy and immunotherapy clinical settings. ”

Regarding the overexpression screen: Although the authors focus on gene pools that accelerate the onset of melanoma, several pools show the opposite effect. In fact, Figure 1B gives the impression that most of the significant pools decrease the onset of melanoma formation. According to the figure legends, 6 pools were significant. The fact that the authors do not list which are the significant ones and do not discuss this in the text is curious. Is there a specific (technical) reason for not looking into genes that delay melanoma onset? Of course, there needs to be focus on one question and for this study, the authors chose to focus on genes with a positive effect on tumor progression, but it would be important to mention which other pools scored significantly and perhaps add some discussion on that. If other genes or pools were validated, it would be important to add that as well.

We have now added a new supplementary figure (now Figure 1—figure supplement 1) showing single factor validation of 4 additional pools (3 positive and a negative pool for a total of 20 single factors). These show other significant accelerator hits from the screen (i.e. TRIM28, CDYL2, DMAP1, CBX5, PYGO2) that had all a much milder phenotype than SATB2. In addition, we show CBX3 as an example of a gene significantly delaying tumor onset compared to the EGFP control.

We have textually edited the Results section as follows:

*“*We identified 6 significant candidate accelerator pools (Pool B,D,F,G,I,L vs. MCR:EGFP Logrank *P*<0.0001****) of which Pool F had the strongest acceleration (Figure 1B), and tested individual factors from 4 accelerating pools (B/F/G/I) and a non significant pool (C) (Figure 1C and Figure 1—figure supplement 1A-D). […] We thus focused on further characterizing SATB2’s phenotype and investigating its mechanism of action.”

We did choose to focus on accelerators over genes delaying tumor onset both for biological interest, but also for various technical reasons. Namely (1) Studying delaying of onset would require a much longer follow up time (2) A screen for genes delaying onset might have more false negatives due to developmental toxicity, particularly since we are testing epigenetic/transcriptional regulators.

We edited the Discussion section to incorporate this point:

“This experimental approach can be used to identify pools/genes that can accelerate (e.g. SATB2 Figure 1C) or delay (e.g. CBX3 Figure 1—figure supplement 1D) tumor onset. […] Of the other 6 significant genes identified in the screen (TRIM28, CDYL2, DMAP1, CBX5, PYGO2, Figure 1—figure supplement 1A-D), TRIM28 is known to increase melanoma tumor propagation potential (Czerwinska et al., 2020).”